# Recent Advances and Applications of Semiconductor Photocatalytic Technology

**Fubao Zhang [1], Xianming Wang [2], Haonan Liu [1,2], Chunli Liu [1], Yong Wan [1,*], Yunze Long [1,*] and Zhongyu Cai [3,4,*]**

1   College of Physics Science, Qingdao University, Qingdao 266071, China; fbzhang18@163.com (F.Z.); liuhaonantongxue@126.com (H.L.); liucl1827@163.com (C.L.)
2   State Key Laboratory of Marine Coatings, Marine Chemical Research Institute Co., Ltd., Qingdao 266071, China; wxm133701@163.com
3   Department of Chemistry, University of Pittsburgh, Pittsburgh, PA 15260, USA
4   Department of Chemical and Biomolecular Engineering, National University of Singapore, 4 Engineering Drive 4, Singapore 117585, Singapore
*   Correspondence: wanyongqd@hotmail.com (Y.W.); yunze.long@163.com (Y.L.); caizhongyu@alumni.nus.edu.sg (Z.C.)

**Abstract:** Along with the development of industry and the improvement of people's living standards, peoples' demand on resources has greatly increased, causing energy crises and environmental pollution. In recent years, photocatalytic technology has shown great potential as a low-cost, environmentally-friendly, and sustainable technology, and it has become a hot research topic. However, current photocatalytic technology cannot meet industrial requirements. The biggest challenge in the industrialization of photocatalyst technology is the development of an ideal photocatalyst, which should possess four features, including a high photocatalytic efficiency, a large specific surface area, a full utilization of sunlight, and recyclability. In this review, starting from the photocatalytic reaction mechanism and the preparation of the photocatalyst, we review the classification of current photocatalysts and the methods for improving photocatalytic performance; we also further discuss the potential industrial usage of photocatalytic technology. This review also aims to provide basic and comprehensive information on the industrialization of photocatalysis technology.

**Keywords:** photocatalysis; application; reaction mechanism; preparation method; catalyst modification; application

## 1. Introduction

As industrialization accelerates, both energy and environmental issues arise. Nowadays, the world's major energy sources are still fossil energies, such as coal, oil, and natural gas. With the continuous development of industrialization, most fossil fuels are expected to be depleted in 21st century. In addition, the use of fossil energy also causes severe pollution to the environment. In recent years, more and more serious air pollution and water pollution directly threaten human life and health [1]. In 1972, Fujishima and Honda found that the $TiO_2$ electrode can break down hydrogen in aquatic production under sunlight [2]. Since then, photocatalysis has attracted intense attention due to its direct conversion of solar energy to easily stored hydrogen, as well as its lack of environmental pollution. Frank and Bard then successfully oxidized $CN^-$ to $OCN^-$ using $TiO_2$ as a photocatalyst, an act which promoted and accelerated the application of photocatalysts in wastewater treatment [3]. Since then, $TiO_2$ has been used as a photocatalyst for a wide range of applications in the field of environmental management. The detailed research of photocatalyst technology also extends the potential application

range of photocatalysts. The main research directions at this stage are photocatalyst disinfection [4–6], photocatalytic hydrogen production [7–11], photocatalytic reduction of $CO_2$ [12–15], photocatalyst wastewater treatment [16–20], and air purification [21–23]. However, photocatalytic technology is only at the laboratory stage, and there is still a long journey to apply this technology in practice [24].

With the continuous development of photocatalytic technology, researchers have published many excellent reviews. For example, Wang et al. [25] reviewed the use of nitrogen doping to modify oxide photocatalysts to improve their photocatalytic performance. Li et al. [26] outlined the application of graphene and its composites to photocatalysts. Zhou et al. [27] gave an overview of the all-solid Z-Scheme photocatalyst system, including the configuration, structure, optimization, and application of a Z-scheme photocatalyst system. Boyjoo et al. [24] presented the application of photocatalysts in air treatment, including catalyst development and reactor design. Wang et al. [28] gave the latest developments and challenges in photocatalytic sterilization. Most of the current reviews focus on only one or several aspects of photocatalysts. A complete summary of the main content of photocatalysts is still missing.

In anticipation of helping researchers with information on more efficient photocatalysts, we have summarized some common preparation and modification methods of photocatalysts. Meanwhile, a series of advances in photocatalyst technology have been reviewed to facilitate researchers' understanding of the latest research trends in photocatalysts. This review article aims to summarize researchers' efforts on the use of various preparation and modification methods to prepare ideal photocatalysts. In addition, this review also promotes the development of photocatalytic technology and even the industrialization of photocatalytic technology.

## 2. Photocatalytic Mechanism and Influencing Factors

### 2.1. Reaction Mechanism

Photocatalytic reaction is a chemical reaction that takes place under the joint action of light and the photocatalyst. This technology possesses several advantages, including environmental protection, the complete degradation of pollutants, and no secondary pollution. Figure 1 shows the basic principle of a photocatalytic reaction [29–40].

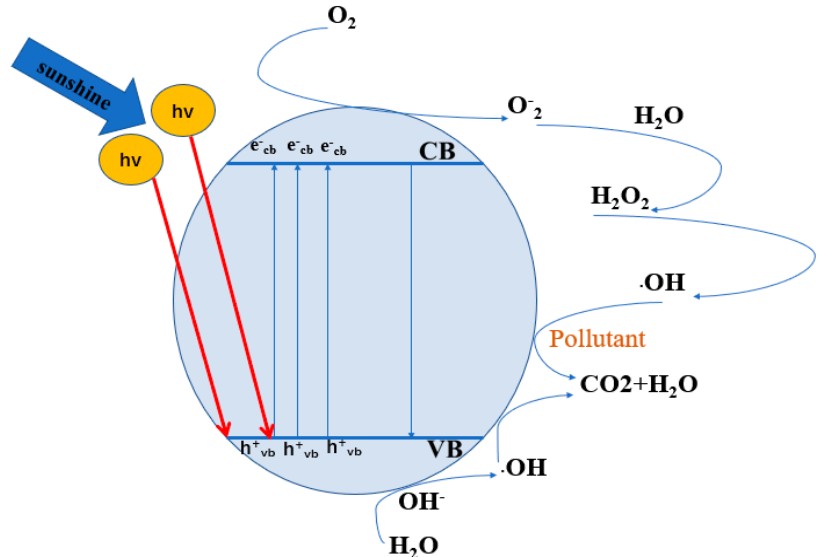

**Figure 1.** Photocatalytic reaction mechanism.

From the point of semiconductor photochemistry, photocatalytic action is the light-induced redox reaction of semiconductors. A semiconductor has an energy band structure composed of a low energy valence band (VB) and a high energy conduction band (CB), and the band gap between a conduction

band and a valence band is called a forbidden band. When the energy of the incident light is larger than the band gap of the semiconductor, the electrons in the VB of the semiconductor are excited to the CB by photons, and the corresponding holes are generated in the VB. Photogenerated electrons and holes are separated by an electric field and move to the surface of semiconductor particles. The photogenerated pores have strong oxidizing properties and can oxidize substances adsorbed on the surface or solution of the semiconductor.

The detailed process of a photocatalytic reaction is as follows. Under a certain energy of light, electrons on a VB are excited and jump to a CB, and the holes stay on the VB. The electrons on the CB move to the surface of the catalyst and participate in a reduction reaction, and the holes on the VB diffuse to the photocatalyst surface and are involved in an oxidation reaction. In the course of the reaction, the electrons can form $H_2O_2$ or a superoxide radical $O_2^-$ with $H^+$ and dissolved $O_2$ in the aqueous solution, and the holes can oxidize $OH^-$ to produce hydroxyl radicals ·OH and, thus, achieve the effect of degrading pollutants [41–44].

## 2.2. Influencing Factors

The photocatalytic reaction process is complicated and has many intermediate stages. Therefore, many factors can influence the photocatalytic reaction [45–55]. The characteristics of the catalyst itself; the surface condition of the catalyst (charge, adsorbed material, defect, composition); the reaction medium conditions (pH, solvent); the type and concentration of reactants, adsorption, and product analysis of reactants; oxygen concentration; light source (wavelength, intensity, distance)—all of these are key factors that affect the photocatalytic reaction. The influence of several factors on the photocatalytic reaction is summarized in Table 1.

**Table 1.** Effects of influencing factors on photocatalytic reaction.

| Influencing Factors | | Effect on Photocatalysis |
|---|---|---|
| Catalyst concentration | | - The reaction rate increases with the increase of the catalyst concentration.<br>- Above a certain dose, the reaction rate decreases as the catalyst concentration increases. |
| Light source and light intensity | Light source<br>Light intensity | - Provide light of different wavelengths.<br>- Improve light intensity and promote photocatalytic reaction. |
| PH value | | - Related to target degradation products. |
| Plus oxidants | | - Reducing the recombination of photogenerated electrons and holes to improve photocatalytic efficiency. |
| Inorganic ion | Anion | - Improve the separation speed of photogenerated electrons and holes and promote photocatalytic reaction.<br>- Becomes a scavenger of hydroxyl radicals, forming anion radicals. |
| | Cation | - The competitive adsorption of active sites on the surface of the catalyst may affect the photocatalytic degradation of organics. |
| Temperature | | - Has little effect. |

## 3. Common Classification of Photocatalysts

### 3.1. Oxide Photocatalyst

#### 3.1.1. TiO$_2$-Based Photocatalyst

In 1976, Carey et al. discovered that nano-TiO$_2$ could dechlorinate polychlorinated biphenyls, a hard-degradable organic compound, under ultraviolet irradiation [18,56–61], and opened a new chapter in photocatalytic degradation of semiconductor nanomaterials. In 1980, Bard et al. proposed a photocatalytic mechanism using TiO$_2$ as a catalyst [62], which promoted the development of TiO$_2$ in the field of photocatalysis.

TiO$_2$ has three crystal forms—rutile, brookite, and anatase. Among these three crystal structures, anatase and rutile-type TiO$_2$ are often used as photocatalysts [63]. TiO$_2$-based photocatalysts have been widely used in air purification, water pollutants degradation, antibacterial disinfection, deodorant, and antifog, among other applications, due to their high activity, stable properties, low cost, and non-toxic environment.

TiO$_2$ has a wide band gap (about 3.2 eV). Therefore, TiO$_2$ can only absorb ultraviolet light when used as a photocatalyst. However, ultraviolet radiation occupies only 5% of sunlight, which greatly limits the application of TiO$_2$ in the visible range. A series of modifications to TiO$_2$ is usually carried out, thereby improving the utilization of visible light by TiO$_2$ and reducing the photogenerated electron-hole recombination rate on the surface of TiO$_2$. In the end, its photocatalytic efficiency can be improved. As shown in Figure 2, common methods used to improve photocatalytic efficiency include semiconductor surface photosensitization, semiconductor surface noble metal deposition, metal/nonmetal doping, and complex semiconductor modification.

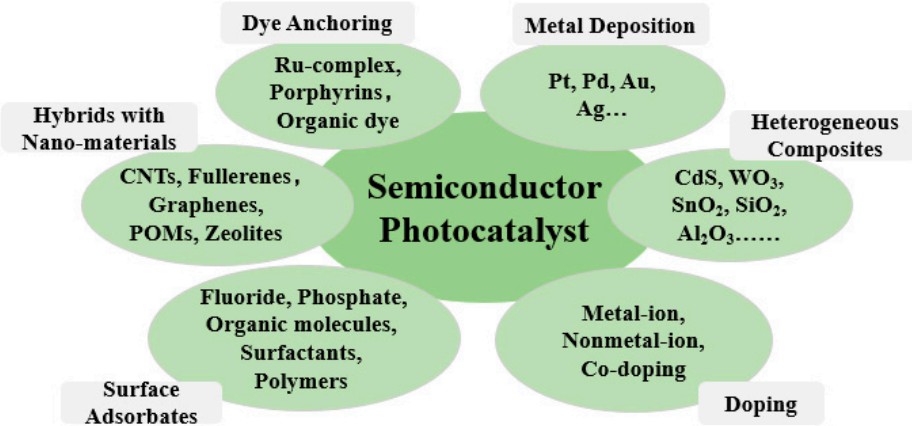

**Figure 2.** Various modification methods of a semiconductor photocatalyst.

### 3.1.2. Bi$_2$O$_3$-Based Photocatalyst

Bi-based oxides are an important class of functional materials that are widely used in many fields [64]. For example, they are used as electronic ceramic materials (zinc oxide varistors, ceramic capacitors, ferrite magnetic materials), electrolyte materials, high-temperature superconducting materials, and materials for photoelectric conversion. Among them, Bi$_2$O$_3$ possesses its own unique properties, such as a high electrical conductivity, low energy bandgap (about 2.8 eV), and low energy band structure, which means that Bi$_2$O$_3$ can absorb visible light in the solar spectrum. Therefore, Bi$_2$O$_3$ has great potential for photocatalysis application [65,66].

The semiconductor Bi$_2$O$_3$ mainly has four crystal forms, including α, β, γ, and δ. The band gaps of β-Bi$_2$O$_3$ and α-Bi$_2$O$_3$ are 2.58 eV and 2.85 eV, respectively, which can absorb visible light with wavelengths exceeding 400 nm [65]. α-Bi$_2$O$_3$ is the most widely used among these four different Bi$_2$O$_3$ semiconductors in photocatalytic reactions, since α-Bi$_2$O$_3$ is the most thermodynamically stable among these four crystal structures [67].

The development of Bi-based semiconductor materials obviously solves the problem of the visible light absorption of TiO$_2$. However, a low quantum efficiency and photogenerated carrier recombination is still an urgent problem that needs to be solved for the Bi-based photocatalysts. The doping modification of the photocatalyst is an effective method for improving its photocatalytic performance. Commonly used modification methods include semiconductor surface photosensitization, semiconductor surface noble metal deposition, metal/nonmetal doping, and complex semiconductor modification. To date, Bi-based photocatalysts have achieved a variety of important research results in the applications of atmospheric purification, organic wastewater treatment, heavy metal ion removal, and sterilization [68–84].

### 3.1.3. Other Oxide Photocatalysts

Transition metal elements of the IV cycle, such as titanium, chromium, manganese, iron, cobalt, nickel, copper, and zinc [85–88], as well as their corresponding oxides, have excellent physical and chemical properties. These excellent physicochemical properties include a high redox potential, multivalence, chemical resistance, and a high temperature resistance. The storage of transition metal elements is relatively abundant and has a low price, which means that transition metal oxides have good application prospects in photocatalysis.

Next, we will introduce the three kinds of transition metal oxides generally used, ZnO, WO$_3$, and Fe$_2$O$_3$.

ZnO is one of the few oxide semiconductor materials that can achieve a quantum size effect, and ZnO has good UV absorption and piezoelectric properties. The forbidden band width of ZnO is 3.2 eV, which corresponds to ultraviolet with a wavelength of 387 nm. Studies on ZnO have found that there is a certain relationship between the surface composition and structure of ZnO nanoparticles and their photocatalytic properties. Oxygen vacancies on the ZnO surface tend to capture photogenerated electrons, and there is a strong interaction between oxygen vacancies and adsorbed oxygen, which is beneficial for the oxidation reactions.

WO$_3$ is one of the metal oxides having photocatalytic activity. WO$_3$ shows several advantages, including a large specific surface area, excellent absorbing capability, and its potential to be used as invisible material. The band gap of WO$_3$ is 2.8 eV, and WO$_3$ is stable. WO$_3$ can be used both as a main catalyst and as a cocatalyst.

Iron has long been investigated as an important meta-element in the redox reaction. Among iron oxides, Fe$_2$O$_3$ has been widely studied for its high photocatalysis activity. Fe$_2$O$_3$ is an n-type semiconductor, having a forbidden band width of 2.2 eV, a strong light absorption capability in the visible light region, and the ability to absorb a part of sunlight. Fe$_2$O$_3$ can be used as a reducing agent for the photocatalytic reduction of silver ions [89,90] and can also be used to decompose reactive dyes [91–93].

### 3.2. Non-Oxide Photocatalyst

Many non-oxides also have good photocatalytic properties [94,95]. Among them, semiconductor metal sulfides have attracted intense attention because of their special structure and excellent physical and chemical properties. Some such non-oxides include as molybdenum sulfide (MoS$_2$), tungsten sulfide (WS), copper sulfide (CuS) [96], zinc sulfide (ZnS) [97,98], and cadmium sulfide (CdS) [99]. Of course, in addition to the sulfides, nitrides, such as C$_3$N$_4$, also show excellent photocatalytic properties.

Next, we will take CdS, CuS, ZnS, and C$_3$N$_4$ as examples to demonstrate their applications in photocatalysis.

### 3.2.1. CdS Series Photocatalyst

CdS is a semiconductor material, having a bandgap of about 2.42 eV and a maximum absorption peak of 514 nm. Therefore, CdS can absorb visible light or ultraviolet light with a wavelength of less than 514 nm, which makes CdS more efficient for visible light photocatalysis [100,101]. In addition, the bandgap position of CdS semiconductors is well suited for many photocatalytic reactions, such as water decomposition [100] and CO$_2$ reduction [102]. More importantly, the position of the CdS conduction band edge is lower than the position of other common semiconductors (such as TiO$_2$, SrTiO$_3$, and ZnO) [103]; this means that in the photocatalytic reaction, photoelectrons of CdS have a stronger reducing power. Therefore, CdS has been intensively investigated as a photocatalyst. However, the CdS material is prone to light corrosion, which results in severe limitations on the number of recoverable photocatalyst. To solve this fundamental problem, researchers have proposed some of measures to improve the use ratio of CdS. The most common method is to prepare CdS composite materials using

different materials or ion doping methods. The CdS composite also enhances the absorption of long wavelength light, thereby achieving the full utilization of the visible part of natural light.

### 3.2.2. CuS Series Photocatalyst

CuS is a very important metal sulfide semiconductor material with excellent optoelectronic properties, and it is considered to be a typical P type semiconductor material. CuS has a forbidden band width of 2.2 eV, which has good visible light absorption capacity. When the CuS material absorbs the proper photons, electron-hole pairs are generated. Electrons and holes move to the surface of the material and react with water molecules or oxygen molecules on the surface of the material to produce several substances. These substances have high catalytic activity and can degrade the organic matter adsorbed on the surface of the material.

Because of its excellent properties, CuS has found a wide range of optical, electrical, mechanical, and sensor applications [104–107], including solar cells, lithium battery electrodes, photothermal conversion, and photocatalysis [108]. CuS nanomaterials have great potential for photocatalysis due to their appropriate energy band and absorption wavelength [109].

### 3.2.3. ZnS Series Photocatalyst

ZnS is a group of II–VI wide band gap semiconductor materials with a band gap between 3.6 and 3.8 eV. ZnS has two different crystal structures—zinc blende and wurtzite. Among them, zinc blende, also known as β-ZnS, can stably exist at low temperatures, while wurtzite, also known as α-ZnS, can exist stably at temperatures above 1024 °C. ZnS is difficult to oxidize and hydrolyze. More importantly, these properties are still present when the size of ZnS is reduced to the nanometer scale. Thus, ZnS nanomaterials exhibit good photocatalytic activity [110]. Furthermore, ZnS is easily fabricated, non-toxic, and widely used. To date, many research groups have successfully synthesized low-dimensional nanomaterials such as ZnS nanoparticles [40,98,101,111–117], nanowires [118–122], nanotubes [123–126], and nanosheets [127,128].

Among the methods for enhancing the photocatalytic activity of ZnS, the following three are the most common approaches. (1) The specific surface area is increased by changing the morphology of ZnS [129–131]. The high specific surface area can increase the active site on the ZnS surface and can increase the contact area between ZnS and the reactant, thereby increasing the photocatalytic activity of ZnS. (2) By altering the electronic properties and band structure of ZnS through doping with other metals and non-metallic elements [132], the absorption rate of ZnS to visible light is improved, thereby improving the photocatalytic efficiency of ZnS. (3) ZnS forms a heterojunction with other semiconductors [133] or a composite structure with noble metals [134] to reduce the electron-hole recombination rate, thereby improving the photocatalytic performance of ZnS.

### 3.2.4. Nitride Series Photocatalyst

In 1989, Liu and Cohen [135] succeeded in replacing the Si atom with the $sp^3$ hybridization of C atoms according to the well-known β-$Si_3N_4$ structure. Four $sp^3$ hybridized N atoms are bonded to the C atom, and three C atoms are bonded to each N atom to form a nearly planar structure of the carbon–nitrogen compound. In 1993, *Science* published one paper on the successful synthesis of $C_3N_4$ crystals with a hardness higher than that of diamonds using laser sputtering technology in the Harvard University laboratory [136,137], which quickly attracted the attention of the global materials community. By 1996, Teter and Hemley [138] believed that carbon nitride might have a structure of α phase, β phase, cubic phase, quasi-cubic phase, and graphite-like phase.

In 2009, Wang et al. [139] reported on a polymer semiconductor material g-$C_3N_4$ consisting entirely of non-metallic elements. g-$C_3N_4$ is a graphite-like layered structure, and g-$C_3N_4$ has an excellent electron/hole transporting ability, a high thermal stability and chemical stability, and a narrow band gap (2.7–2.8 eV). The forbidden bandwidth of g-$C_3N_4$ allows it to have a strong light absorption in the visible light region (400–450 nm). Furthermore, the structure and properties of g-$C_3N_4$

are easy to control, making it a hot research topic in the field of photocatalysts [140,141]. g-$C_3N_4$ shows excellent photocatalytic performance in the field of photo hydrolysis, hydrogen production, photodegradation of environmental pollutants, $CO_2$ reduction, sterilization, and synthesis of composite capacitor materials [139,141–149].

The forbidden band width of semiconductors is an important factor limiting the wide application of photocatalysts.; as such, the forbidden band widths of some commonly used photocatalysts are summarized in Table 2.

**Table 2.** Band gaps of common photocatalysts.

| Semiconductor | Crystal Structure | Band Gap Structure (PH = 7) | | | Reference |
|---|---|---|---|---|---|
| | | CB | VB | $E_g$/eV | |
| $TiO_2$ | Anatase | −0.50 | 2.70 | 3.20 | [149] |
| ZnO | | −0.31 | 2.89 | 3.20 | [150] |
| CuO | | −1.16 | 0.85 | 2.00 | [151] |
| CdS | | −0.90 | 1.50 | 2.40 | [152] |
| ZnS | | −1.04 | 2.56 | 3.60 | [101] |
| g-$C_3N_4$ | | −1.30 | 1.40 | 2.70 | [139,153] |
| g-$C_3N_4$ | | −1.53 | 1.16 | 2.70 | [154] [a] |
| $Ta_3N_5$ | | −0.75 | 1.35 | 2.10 | [155] |
| TaON | | −0.75 | 1.75 | 2.50 | [156] |
| $Fe_2O_3$ | | 0.28 | 2.48 | 2.20 | [157] |
| $Bi_2O_3$ | | 0.33 | 3.13 | 2.80 | [158] |
| $BiVO_4$ | | −0.30 | 2.10 | 2.40 | [159] |
| $WO_3$ | | −0.10 | 2.70 | 2.80 | [160] |
| $Ag_3PO_4$ | Cubic | 0.04 | 2.49 | 2.45 | [161] |

[a] Measurement by the valence band X-ray photoelectron spectroscopy (VB XPS) spectrum.

## 4. Common Preparation Methods

There are many ways to prepare a photocatalyst. In this review, we divide these methods into the electrospinning method, the solid phase method, the liquid phase method, and the vapor phase method.

Next, we will describe the preparation method of a photocatalyst in detail according to the structure of Figure 3.

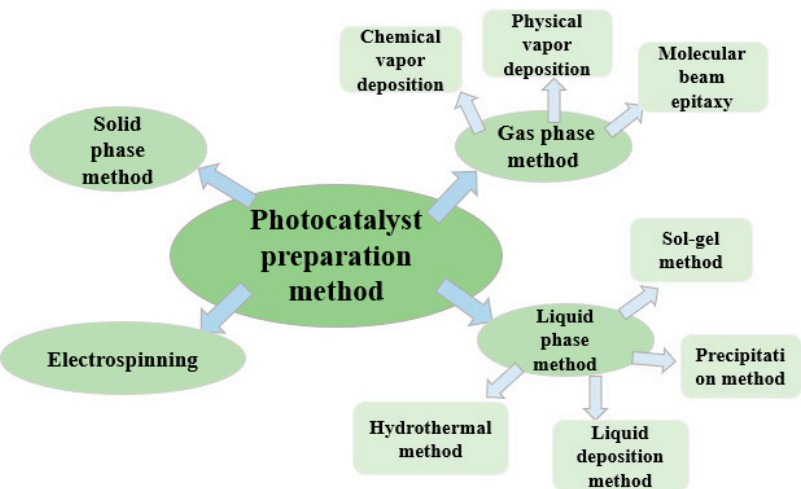

**Figure 3.** Photocatalyst preparation method.

### 4.1. Electrospinning

Electrospinning technology is a method of directly producing polymer nanofibers [161–177]. Electrospinning techniques have the advantages of simple manufacturing equipment, low spinning

cost, a wide variety of spinnable materials, and a controllable process. Electrospinning technology is one of the major methods for effectively preparing nanofiber materials. Electrospinning technology has been used to produce various nanofibers including organic, organic/inorganic composites, and inorganic nanofibers.

Electrospinning device mainly consists of a high-voltage power supply, a nozzle and liquid supply device, and a fiber receiving device, as shown in Figure 4A. High voltage power supplies typically use a DC power supply that can generate thousands to tens of thousands of volts. The function of the high voltage power supply is to generate a high voltage electric field so that the liquid becomes charged and polarized, eventually forming a jet. The liquid supply device is a container (such as a syringe) having a capillary at one end, and the container is filled with a polymer solution or melt. In addition, as experimental requirements have increased, many experiments have gradually adopted liquid flow control systems that allow for a more accurate control of liquid flow. The fiber receiving device is a metal receiving plate at the opposite end of the nozzle and may be a rotating roller or an aluminum foil layer on the plane of the metal plate. The receiving device is grounded with a wire and connected to the negative electrode of the high voltage power source.

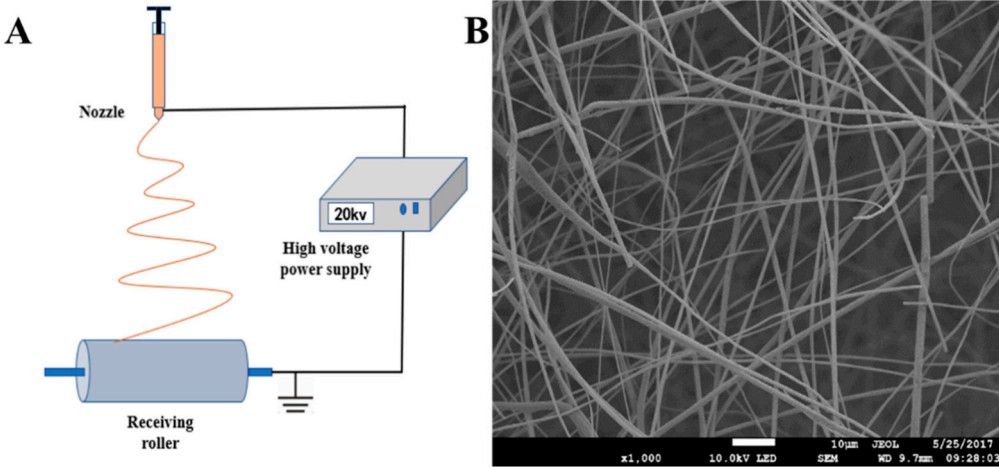

**Figure 4.** (**A**) Schematic illustration of electrospinning and (**B**) SEM image of TiO$_2$ nanofibers prepared by electrospinning.

During electrospinning, the high voltage electric field melts and deforms the polymer solution, which then forms conical droplets that protrude at the end of the showerhead. When the repulsive force of the surface of the droplet exceeds the surface tension, a minute jet ejects from the surface of the droplet. These jets undergo a high-speed stretching of electric field force at short distances, the solvent volatilizes and solidifies, and, finally, it deposits on the receiving plate to form polymer fibers. Figure 4B shows an SEM image of a polyvinylidene fluoride (PVDF) /TiO$_2$ fiber film prepared by an electrospinning method. We can observe that the nanofiber membrane prepared by electrospinning has a porous structure. The conventional fiber has a specific surface area of 0.4 m$^2$g$^{-1}$, as compared with a conventional fiber membrane, and the electrospinning fiber has a specific surface area of about 40 m$^2$g$^{-1}$ [163]. The electrospinning fiber membrane has a large specific surface area, which can effectively solve the disadvantage that the specific surface area of the photocatalyst is small. In addition, the prepared fiber membrane is advantageous for the recovery of the photocatalyst and the reduction of secondary pollution.

In the process of manufacturing nanofibers by electrospinning, the voltage, solution viscosity, surface tension, solvent evaporation rate, and conductivity of the electrospinning operation are numerous, and their relationship is complex. These factors all affect the average diameter and uniformity of the fiber to different extents.

### 4.2. Solid Phase Method

The solid phase method is a method in which a reaction raw material is thoroughly mixed and ground at a certain stoichiometric ratio before being calcined at a specific temperature to obtain a product [178–182]. The solid phase method has the advantages of a simple apparatus, convenient operation, low cost, uniform particle size, and controllable force. In the solid phase method, no solvent is required; thus, hard agglomeration, which may occur in the liquid phase, can be avoided, and environmental pollution can be reduced. However, the solid phase method also has the disadvantage that the particles are easy to aggregate, the powder is not fine enough, the impurities are easily mixed, and the oxidation of the ions is easy.

### 4.3. Gas Phase Method

The gas-phase method refers to direct use of gas or a means to change a substance into a gas—making it physically or chemically react in a gas state—to agglomerate and grow to form nanoparticles in the cooling process. In the gas phase method, nanoparticles having high purity, and good nanoparticle properties can be produced. However, the gas phase method also requires advanced techniques and equipment. Vapor phase methods include physical vapor deposition (PVD), chemical vapor deposition (CVD), and molecular beam epitaxy (MBE).

#### 4.3.1. Chemical Vapor Deposition (CVD)

Chemical vapor deposition (CVD) [183,184] is a method of forming a thin film by chemical reaction to one or more vapor phase elements or compounds on the surface of a substrate. CVD can be used to deposit metals, carbides, nitrides, oxides, borides, and the like. CVD can be applied to the surfaces of complex geometric shapes and has a good adhesion to the film base. Therefore, the development of CVD is very rapid. Depending on different reactants and control conditions, CVD can be further divided into atmospheric pressure chemical vapor deposition, low pressure chemical vapor deposition, plasma CVD, laser CVD, and metal–organic CVD. Thanks to the development of thin film manufacturing technology, CVD has been continuously developed and improved. However, the equipment and cost required for CVD is relatively high. In addition, CVD possesses many disadvantages, such as uneven microscopic surface roughness of the resulting film and particle size (50–150 nm), as well as a relatively large surface roughness of the film.

#### 4.3.2. Physical Vapor Deposition (PVD)

Physical vapor deposition (PVD) is a method in which a paint is vaporized by a physical method to form a film on the substrate surface. In addition to conventional vacuum deposition and sputter deposition techniques, PVD includes various ion beam deposition, ion plating, and ion beam assisted deposition techniques that have prospered in recent years. Types of deposition include vacuum deposition, sputtering, and ion plating [185–188]. PVD is a common technique for producing hard coatings (hard films). Compared with CVD, the PVD method has a low deposition temperature and does not easily cause deformation and cracking of the substrate. Films produced by PVD are uniform and possess easy to control structures and properties. However, the PVD preparation of the film must be carried out under a vacuum, and the equipment necessary for film formation is expensive.

To date, PVD methods can deposit not only alloy films and metal films but also ceramics, compounds, polymer films, semiconductors, and so on.

#### 4.3.3. Molecular Beam Epitaxy (MBE)

Molecular beam epitaxy (MBE) is a newly developed method of film production and is also a special vacuum coating process. MBE is a new manufacturing technology for single crystal films [189–193]. MBE means to spray a hot atomic beam or molecular beam onto the surface of a heated substrate under vacuum conditions and react with the surface of the substrate to deposit the film.

The advantages of MBE include the facts that the beam intensity is easy to control precisely, the surface morphology of the epitaxial material can be made into a multi-layer structure of different compositions, and the composition and doping of the epitaxial layer can be changed at will. MBE can accurately control the thickness of the epitaxial layer, the doping profile of the heterojunction interface, and the flatness of the heterojunction interface, which is beneficial for improving the purity and integrity of the epitaxial layer. MBE has made it possible to control the precise growth of atomic layers and subsequent atomic layers. MBE can produce thin monocrystalline thin films on the order of tens of atomic layers and can also produce thin films with different doping and different compositions.

## 4.4. Liquid Phase Method

The liquid phase method is also called a wet chemical method. In the liquid phase method, it is necessary to first select an appropriate soluble metal salt and then prepare a solution according to the composition of the prepared material [194–196]. Second, suitable precipitating agents are selected to precipitate or crystallize the metal ions in the solution; they may also operate by evaporation, sublimation, hydrolysis, etc. to cause precipitation or crystallization. Finally, the precipitate or crystal is dehydrated or thermally decomposed to obtain the desired raw material powder.

There are many methods for preparing photocatalytic films by liquid-phase method, such as the sol–gel method, the precipitation method, the liquid deposition method, and the hydrothermal method.

### 4.4.1. Sol–Gel Method

The sol–gel method is a wet chemical approach used to prepare nanoparticles [197]. In the sol–gel method, a precursor is dissolved in a solvent, and a sol is formed through hydrolysis or alcoholysis. The sol is then converted into a gel after prolonged storage or drying. Figure 5 shows the process of preparing nanomaterials by a sol–gel method.

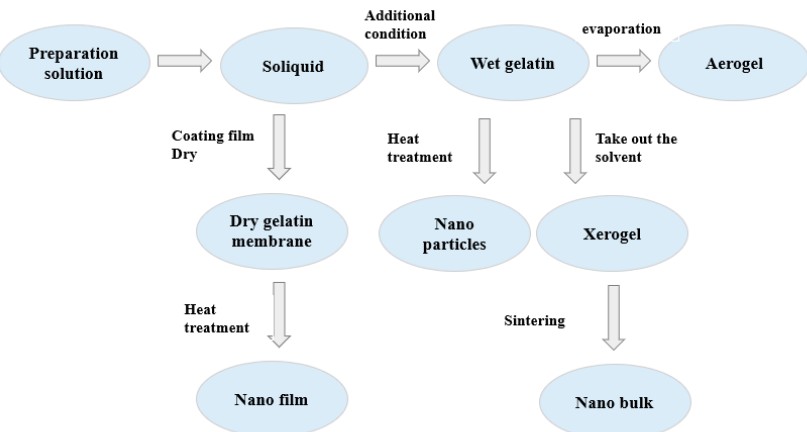

**Figure 5.** Nanofilm or nanoparticles by a sol–gel method.

Among the nanoparticles prepared by the sol–gel method, the most important factors are the formation of the sol and gel. There are many complicated parameters affecting the sol–gel process. At present, most people think that the four main parameters have important influence on the sol–gel process. These are the pH of the solution, the concentration of the solution, the reaction temperature, and the reaction time.

The sol–gel process has the advantages of high purity, high uniformity, low synthesis temperature, and easy control of the reaction conditions [198–200]. The preparation process of the sol–gel process is relatively simple and does not require special or expensive equipment. The problem of the sol–gel method is that the raw material is generally titanium alkoxide, a large amount of organic solvent is required, and the obtained film is heat-treated at a relatively high temperature. Therefore, the film forming cost is relatively high, the adhesion of the film is poor, and the transparency is poor.

### 4.4.2. Precipitation Method

The precipitation method is one of the most common methods for synthesizing nanomaterials. In the precipitation method, substances with different chemical components are mixed, and a precipitant is added to obtain a precursor precipitate. The precursor precipitate is then dried or calcined to produce the corresponding nanoparticles [201–206]. When the particle diameter is about 1 μm, a precipitate is formed. The particle size of the particles generally depends on the solubility of the precipitate and the supersaturation degree of the solution. The smaller the solubility of the precipitate is, the smaller the diameter of the particle produces, and the smaller the supersaturation degree of the solution is, the larger the diameter of the particle forms.

The photocatalyst prepared by the precipitation method has good light absorption capability and can promote the separation of holes, and it can thereby improve photocatalytic performance. In the precipitation process, many of the heavy metals added do not readily react in solution and may introduce impurities into the solution, which limits the application of the precipitation method. During the operation of the precipitation method, the reaction temperature, time, pH value, ratio of reactants, titration rate, and other factors have a certain influence on the material preparation. The precipitation method shows the characteristics of low cost, simple process, and safety.

### 4.4.3. Liquid Deposition Method

In the liquid deposition method, a substrate is immersed in an appropriate reaction solution, and a uniform and dense film is deposited onto the substrate. These films obtained by the liquid deposition process usually consist of oxides or hydroxides [207–211]. The liquid deposition method is suitable for substrates of various shapes, and it does not require a high temperature during film formation or expensive equipment. The liquid deposition method can be used not only to prepare a single oxide film but also to prepare a composite oxide film, a multi-component oxide film, a metal fine-particle-dispersed oxide film, a laminated oxide film, and so on. The reaction principle of liquid deposition method is not complicated. The process is simple, and the film formation rate is high. The benefit/cost ratio is large, while the pollution to the environment is small. In addition, several disadvantages of liquid deposition methods have been solved, which means that liquid deposition methods have wide application prospects.

For example, Zhou et al. [212–214] immersed a substrate glass sheet in a reaction solution to obtain a transparent anatase type $TiO_2$ film at 35 °C. The reaction solution containing a complex $TiF_6^{2-}$, $F^-$ ion trapping agent $H_3BO_3$ and a crystallization-inducing agent of $TiO_2$. The thickness of the film obtained by the liquid phase deposition method increases as the deposition time increases, and when the deposition time is 9 h, the resulting film thickness is about 260 nm. The photocatalytic activity of $TiO_2$ thin film after heat treatment at different temperatures was evaluated by the photocatalytic degradation of a methylene blue experiment. The results showed that the $TiO_2$ film heated at 300 °C has the highest photocatalytic activity, and its activity is equivalent to five times that of $TiO_2$ film deposited at 35 °C.

### 4.4.4. Hydrothermal Method

In the hydrothermal method, an inorganic or organic compound is mixed with water at a temperature of 100–350 °C and a high pressure, and then an improved inorganic substance is obtained through several steps. The resulting inorganic substance is filtered, washed, and dried to obtain ultrafine particles of high purity [7,215–220].

The hydrothermal preparation of the film is carried out in the liquid phase, and no post treatment is required. This feature of the hydrothermal method avoids defects such as curl, cracks, particle coarsening, and film-to-gas reactions that can occur during heat treatment of the film. The hydrothermal method uses an inorganic substance as the precursor and water as the reaction medium, which avoids a series of disadvantages resulting from the use of organometallic materials. The hydrothermal method

has many advantages, including mild conditions, a stable system, little environmental pollution, an easy process, and a low cost. In addition, films prepared by the hydrothermal method have good uniformity, firmly adhere to the substrate, and are not limited by the shape and size of the substrate [221]. The hydrothermal method can also achieve uniform doping and preparation of the nanocomposite by adjusting the proportion of the product or by adding a surfactant.

## 5. Methods for Improving Photocatalytic Efficiency

By investigating the mechanism of photocatalytic reaction, it is known that semiconductor photocatalysis shows the advantages of mild reaction conditions, high efficiency, and a wide application range. However, semiconductor photocatalytic reactions also suffer from drawbacks in practical applications such as narrow absorption spectra, low photon quantum efficiency, and an easy recombination of photogenerated holes. The recombination of photogenerated electrons and holes is unfavorable for photocatalytic reactions. Therefore, reducing the recombination of photogenerated electrons and holes is an effective way to improve photocatalytic efficiency [48,222,223]. Many methods have been reported for improving the photocatalytic efficiency of semiconductors such as precious metal deposition, semiconductor compounds, metal or non-metal particle doping, and surface dye sensitization.

### 5.1. Precious Metal Depositing

Precious metal deposition on semiconductor surfaces considered to be an effective way of capturing excited electrons. The basic principle of precious metal deposition on a semiconductor surface is as follows. When the noble metal and the semiconductor are in contact, the work function of the metal is larger than the work function of the semiconductor, so the electron continuously moves from the semiconductor to the metal until the energy levels of both are equal. Thus, the metal surface and the semiconductor surface each obtain excessive negative and positive charges, causing band bending near the interface where the semiconductor is in contact with the metal, thereby forming a Schottky barrier. The Schottky barrier can capture photogenerated electrons at the surface of the semiconductor and suppress the recombination of electron-hole pairs [224], thereby improving photocatalytic efficiency.

Among the surface precious metal deposits, the most commonly used metal is the Pt of Group VIII [225–232]. In addition to Pt, noble metals such as Ag [233–238], Ru [239,240], Pd [241–245], and Au [246–250] are also used to improve the photocatalytic performance of semiconductors. However, the major disadvantage of precious metals such as Ag, Ru, Pd, and Au is that the price is relatively high and not suitable for large scale applications [251–257].

Take $TiO_2$ as an example. Due to the large ionic radius of the noble metal, it is impossible to enter the $TiO_2$ lattice. Therefore, in order to improve the photocatalytic performance of $TiO_2$, it is possible to change the surface characteristics of $TiO_2$ by depositing a noble metal on the surface of $TiO_2$ photocatalyst [258–261]. The surface modification of a noble metal to $TiO_2$ photocatalyst is achieved by changing the electron distribution [261]. When $TiO_2$ contacts the noble metal, carriers are redistributed, and a space charge layer is formed on the surface of the $TiO_2$ close to the metal particle interface. The space charge layer facilitates the transfer of photogenerated electrons to the interface between the noble metal and $TiO_2$ [258,259,262], thereby suppressing the recombination of photogenerated electrons and holes, improving photocatalytic efficiency.

Klein et al. [263] deposited single atom and diatomic clusters (Pd, Pt, Ag/Pd, Ag/Pt, and Pd/Pt) on the surface of $TiO_2$ (P25) using a high energy radiation reduction method. By the photocatalytic decomposition of toluene, it was found that palladium/platinum–titanium dioxide (Pd/Pt–$TiO_2$) has the strongest decomposing ability for toluene in the ultraviolet and visible range. Two methods in Klein's experiment are shown in Figure 6. In the comparison of different preparation methods, it was found that the sample prepared by the one-step method had the highest degradation ability to phenol in the visible range. Zhang et al. [264] deposited Au nanoparticles on the surface of sulfated $TiO_2$. Au nanoparticles have excellent electron trapping ability and can effectively prolong the

lifetime of electron-hole pairs. In the photocatalyst experiment, under the condition of pH = 6.5, the decomposition rate of the photocatalyst to Congo red dye can reach 97.6%. Padilla et al. [265] used an innovative sonic degradation deposition method to deposit a series of noble metals on the surface of $TiO_2$ within 10–20 s. These precious metals include single or complex noble metals such as Au, Cu, Pd, Pt, Au–Pd, Au–Cu, Au–Fe, Au–Co, Au–Pt–Au–Rh, and Au–Ru. In photocatalytic experiments under room temperature and visible light conditions, the decomposition rate of methyl orange dye was highest with the AuPd/$TiO_2$ (mass ratio of Au: Pd is 3:1) catalyst.

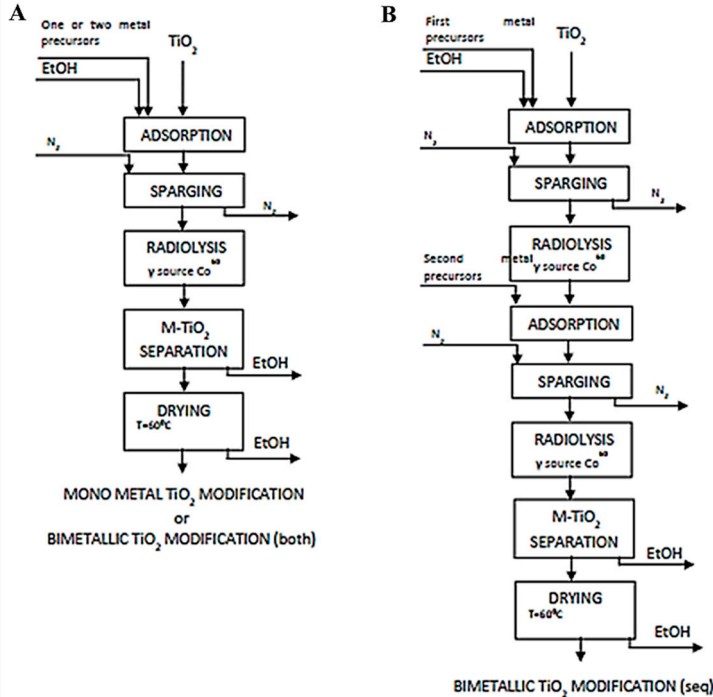

**Figure 6.** Block diagram of (**A**) mono metal or bimetallic (both) $TiO_2$ modification and (**B**) bimetallic $TiO_2$ modification (seq) [263].

Most precious metals are toxic, expensive, and not suitable for a wide range of applications. Relatively speaking, Ag is one of the most popular precious metals because it is less toxic and not expensive. Jaafar et al. [266] applied a mild in situ electrochemical method to deposit Ag nanoparticles on the surface of $TiO_2$. A photocatalytic decomposition experiment of chlorophenol was carried out using the prepared catalyst under the condition of pH = 5. After 6 h of photocatalytic decomposition, the decomposition rate of the Ag–$TiO_2$ photocatalyst was 94%, much higher than that of the other photocatalysts. Though the silver–oxygen–titanium (Ag–O–Ti) chemical bond will reduce the catalytic activity of the material to some extent, the proper content of Ag and the oxygen vacancies on the surface of the material can effectively prevent the recombination of electron-hole pairs, thereby improving the photocatalytic efficiency of the photocatalyst.

*5.2. Semiconductor Compound*

A semiconductor composite photocatalyst is realized by two types of band-matched semiconductors. The methods for achieving semiconductor recombination include simple combinations, doping, multilayer structure, and out of phase combination. The essence of semiconductor blending to improve the photocatalytic performance is that photogenerated electrons or holes generated by a semiconductor move to the conduction band or the valence band of another semiconductor to separate photogenerated electrons and holes, thus effectively suppressing the recombination of carriers [252].

Arabzadeh et al. [267] combined a single-dimensional cadmium sulfide (CdS) nanowire with a $TiO_2$ nanoparticle by a hot solvent method to form a catalyst for a core-shell structure. In photocatalysis

experiments, cadmium sulfide/titanium dioxide ($CdS/TiO_2$) photocatalysts can completely decompose methyl orange, methylene blue, and rhodamine B, and the photocatalytic decomposition time is only 2–3 min. Wu et al. [268] prepared a nanorod $FeVO_4$–$TiO_2$ composite catalyst by the coprecipitation method. In the NO gas conversion experiment, the decomposition rate of NO by the nanorod $FeVO_4$–$TiO_2$ composite catalyst was more than 1.5 times that of the nanoparticle $FeVO_4$–$TiO_2$ composite catalyst. Chu et al. [269] prepared a $MoSe_2$–$TiO_2$ composite photocatalyst by using a hot solvent method. $Cr^{6+}$ ions were reduced by using pure $MoSe_2$, pure $TiO_2$, and complex $MoSe_2$–$TiO_2$ as the photocatalysts. After 120 min of visible light irradiation, the reduction rate of $Cr^{6+}$ by pure $MoSe_2$ and pure $TiO_2$ was only 61% and 2%, respectively, while the conversion rate of composite catalyst $MoSe_2$–$TiO_2$ was as high as 91%. Das et al. [270] prepared a zirconium dioxide–titanium dioxide ($ZrO_2$–$TiO_2$) composite semiconductor photocatalyst by the sol–gel method. In the research of industrial printing and dyeing wastewater purification, there is a relationship between the degradation rate of pollutants and the radiation time, pH, sewage concentration, and catalyst concentration. The concentration of organic pollutants, especially, has the greatest influence on the degradation rate of pollutants.

Shen et al. [271,272] reported the use of wet chemical methods to synthesize large-sized porous $InVO_4$ spheroids and large-sized porous g-$C_3N_4$ particles. Next, nano-$TiO_2$ particles were composited on the surfaces of $InVO_4$ and g-$C_3N_4$ to obtain a semiconductor composite material having a large size, porosity, and narrow band gap. Experiments show that, in addition to improving the quantum effect of the material, the semiconductor composite material also increases the specific surface area of the material. Higher specific surface areas provide more reactive sites, thereby extending the lifetime of photogenerated electron-hole pairs.

In multicomponent modified $TiO_2$, as compared to one-component modified $TiO_2$, different components can be combined with one another such that the catalyst has different degrees of improvement in light absorption performance and electron-hole separation efficiency. Gao et al. [273] used photoreduction deposition and chemical bath deposition to prepare Pt and CdS nanoparticles on co-modified $TiO_2$ nanotubes. The decomposition rate of methyl orange dye with Pt and CdS co-modified $TiO_2$ nanotubes reached 91.9% under visible light. The decomposition rate of methyl orange fueled by a pure $TiO_2$, $Pt/TiO_2$, and $CdS/TiO_2$ single composite catalyst is much lower than 91.9%. Experimental results showed that CdS and Pt have a synergistic effect in which CdS can attract holes to form an active center and Pt can promote photogenerated electron transfer. Therefore, $TiO_2$ co-doped with CdS and Pt has a significant improvement in photon capture rate and carrier separation efficiency. Feng et al. [274] used a hydrothermal method to prepare a multilayer semiconductor composite $TiO_2$ nanotube array. Under the irradiation of solar light, the composite $TiO_2$ nanotube can completely degrade p-nitrophenol and rhodamine B in 20 min and 80 min, respectively.

*5.3. Metal or Non-Metal Ion Doping*

The introduction of impurity in the forbidden band by ion doping is one of the most common and effective methods for improving semiconductor performance. As early as 1982, Borgarello [275] mixed chromium ions ($Cr^{5+}$) into $TiO_2$ to realize visible light cracking water. On one hand, ion doping can increase the concentration of semiconductor carriers, but, on the other hand, it is possible to form ion traps to capture electrons and holes and reduce the recombination of electron-hole pairs. Ion doping causes the red shift of the $TiO_2$ photocatalyst, and the photo response of the photocatalyst can be extended to the visible region. According to the types of doped ions, ion doping can be classified into metal ion doping, non-metal ion doping, and metal/non-metal ion simultaneous doping.

5.3.1. Metal Ion Doping

Metal ion doping is the most commonly used method for photocatalyst modification. The metal ion doping method is to dope transition metal ions in the catalyst by high-temperature firing or auxiliary deposition. The influence of metal ion doping on the photocatalyst mainly includes the

suppression of recombination of photogenerated electrons and holes, as well as the expansion of the response spectrum range of the photocatalyst. By doping $TiO_2$ with chromium and vanadium ions, the excitation wavelength range of $TiO_2$ can be expanded to the visible light region (to the vicinity of 600 nm). However, the incorporation of metal ions sometimes becomes the recombination center of electrons and holes, affecting the photocatalytic effect. The type and concentration of doping ions are different, and the influence on the photocatalysis experiment also differs. Experimentally doped metal ions include transition metal ions and rare earth metal ions.

Transition metal ions mainly include cobalt ions ($Co^{2+}$), iron ions ($Fe^{3+}$), and copper ions ($Cu^{2+}$). Different metal ion doping shows different effects on the performance of a photocatalyst. Doping some specific metal ions can improve the photon efficiency, such as $Fe^{3+}$ [255], while doping some other metal ions, such as $Cr^{3+}$, is harmful [253]. Choi et al. [276] studied more than 20 metal ion doped $TiO_2$ nanocrystals and found that doping 0.5 wt% of $Fe^{3+}$, $Ru^{2+}$, $Mo^{5+}$, $Re^{2+}$, $Os^{2+}$, $V^{5+}$, and $Rh^{2+}$ in the crystal lattice can increases the photocatalytic activity of photocatalyst. Wang et al. [277] applied a simple hydrothermal method to prepare $Fe^{3+}$ doped $TiO_2$ nanotube array catalysts. The semiconductor catalyst prepared at a concentration of $Fe^{3+}$ of 1 mmol/L showed good catalytic performance under irradiation with a solar lamp. The time required for the $Fe^{3+}$ doped semiconductor catalyst to decompose 98.79% methyl blue is only 120 min. The reduction ratio of $Cr^{6+}$ by the $Fe^{3+}$-doped semiconductor catalyst is as high as 83.79%, which is much higher than the reduction ratio of pure $TiO_2$ to $Cr^{6+}$ (39.6%). Inturi et al. [278] produced $Cr^{3+}$ doped $TiO_2$ nanoparticle photocatalysts by three preparation methods and found that the doping samples prepared by flame spraying method have higher catalytic performance than other preparation methods. The sample prepared by the flame spray method degraded chlorophenol under visible light, and the decomposition rate reached 61% in just 5 h. Nanoparticles prepared by the flame spray method have a high specific surface area and increase the active site of the complex, thereby improving its photocatalytic efficiency. Ma et al. [279] used a co-precipitation method to prepare Mg, Zn, and Cr—three kinds of ion co-doped $TiO_2$ nano-catalysts—and used them to degrade Congo red dye. Experimental results showed that the MgZnCr–$TiO_2$ catalyst has good catalytic performance in the visible range. The MgZnCr–$TiO_2$ sample has recyclability and maintains a stable photocatalytic performance after five cycles.

Rare earth elements were originally used as surface treatment agents for $TiO_2$ to improve the vividness of $TiO_2$. With the rapid development of photocatalysts, the role of rare earth elements becomes more important. Rare earth metal ions have a special electronic layer structure compared to transition metal ions. In addition to the benefits of doped transition metals, the doping of rare earth metal ions can also cause semiconductor lattice distortion and impurity defects, form traps for electron holes, and enhance the quantum effect of the semiconductor catalyst. Villabonaleal et al. [280] analyzed the catalytic properties of Lanthanum (La), Cerium (Ce), Praseodymium (Pr), Neodymium (Nd), Samarium (Sm), Europium (Eu), and Gadolinium (Gd) rare earth doped $TiO_2$. They found that the atomic number and concentration of the lanthanide influence the band gap energy and the specific surface area of the catalyst but have little influence on visible light absorption. Malengreaux et al. [281] prepared $TiO_2$ semiconductor photocatalysts with $Fe^{3+}$, $Cr^{3+}$, $La^{3+}$, and $Eu^{3+}$ ions doped and co-doped by a sol–gel method. In the experiment of decomposing p-nitrophenol under ultraviolet and visible light, the La–Fe–$TiO_2$ catalyst showed better photocatalytic ability than the other samples.

### 5.3.2. Non-Metal Ion Doping

The doping of many metals such as $V^{3+}$, $V^{4+}$, $Cr^{3+}$, $Mn^{3+}$, $Fe^{3+}$, $Co^{3+}$, $Ni^{2+}$, $Zn^{2+}$, $Ga^{3+}$, $Zr^{4+}$, and $Nb^{5+}$ can expand the spectral response range of $TiO_2$ to the visible region [282,283]. In addition, some metal-doped $TiO_2$ are not thermodynamically stable and easily cause an increase in recombination centers, thereby reducing the light absorption efficiency of $TiO_2$. Many researchers have therefore chosen to dope $TiO_2$ with non-metal atoms that form covalent bonds with titanium atoms [284]. In 2001, Japanese scientist Asahi et al. [285] prepared a $TiO_{2-x}N_x$ photocatalyst using non-metallic N-doped $TiO_2$. The $TiO_{2-x}N_x$ photocatalyst extends the optical excitation wavelength of $TiO_2$ to the visible region

of 400–520 nm (Figure 7A) and maintains the photocatalytic activity of $TiO_2$ in the ultraviolet region. Methyl blue and acetaldehyde decomposition experiments showed that the $TiO_{2-x}N_x$ photocatalyst shows high photocatalytic activity under both ultraviolet and visible light (Figure 7B).

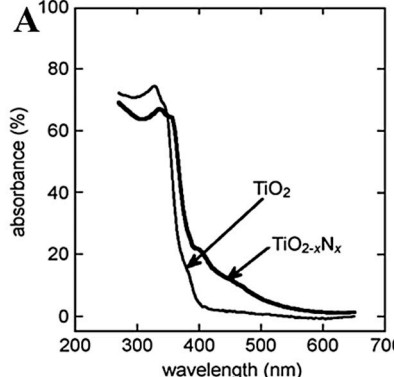
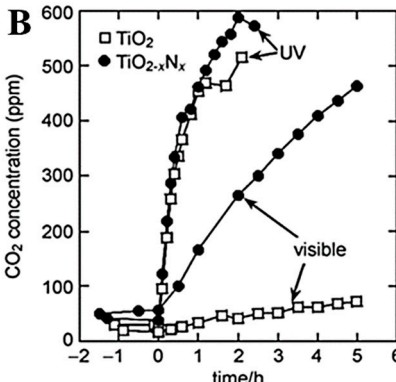

**Figure 7.** (**A**) UV-Vis spectra of $TiO_{2-x}N_x$ and $TiO_2$ and (**B**) the photocatalytic property of acetaldehyde [285].

The research of $TiO_2$ has opened a new research area on a non-metal doped modified $TiO_2$ visible light catalyst, which is undoubtedly a pioneering work. Afterwards, scientists conducted systematic and detailed research on the doping modification of a $TiO_2$ photocatalyst with nonmetallic elements. Non-metals generally used for semiconductor doping are N [285–290], C [55,291–293], S [101,292,294–299], P [297,300,301], F [290,302–307], Cl [308], I [309], B [310–313], and Si [314–316], among others.

Kitano et al. [286] reported the preparation of N–$TiO_2$ films by the radio frequency (RF) magnetron sputtering (RF-MS) deposition method, in which an $N_2$/Ar mixture was used as a sputtering gas. It was found that the absorption band of the N–$TiO_2$ photocatalyst was shifted to the visible region, and the catalytic performance of the N–$TiO_2$ photocatalyst under visible light was remarkably improved. Simsek et al. [317] described the use of a hot solvent method to prepare a boron (B) ion doped $TiO_2$ catalyst, and they studied the effect of the boron doping amount and pH on the degradation of environmental pollutants. The experiment result shows the following: After UV irradiation for 300 min, the decomposition rates of pure $TiO_2$ and B–$TiO_2$ relative to dichlorophenol were 75.7% and 89.7%, respectively, and the decomposition rates of bisphenol A were 97.3% and 99.1%, respectively. After irradiation with visible light for 300 min, the decomposition rates of pure $TiO_2$ and B–$TiO_2$ relative to dichlorophenol were 20.2% and 75.1%, respectively, and the decomposition rates of bisphenol A were 46.0% and 97.7%, respectively. Since the forbidden band width of the B-doped $TiO_2$ semiconductor decreases, the modified material has an absorption response in the visible light region. In addition, two chemical bonds of B–O–Ti and B–O–B are formed during the doping process, and these two chemical bonds not only provide stable performance but also function as traps of photogenerated electrons. He et al. [318] synthesized a C-doped $TiO_2$ photocatalyst by the hydrothermal method. As a carbon source, chitosan forms a porous morphology of the composite $TiO_2$, and a Ti–C chemical bond in the complex that promotes formation of surface oxygen vacancies. The porous morphology of the composite $TiO_2$ and the surface oxygen vacancies together improve the capture rate of organic molecules by the photocatalyst and improve the photocatalytic performance. The conversion rate of composite $TiO_2$ under visible light to NO gas can reach 71%.

Multi-component non-metal doping can also improve the catalytic ability of $TiO_2$ to some extent. Elsheikh et al. [319] used a one-step hydrothermal method to synthesize C and N element co-doped $TiO_2$ photocatalysts, and they used different molar concentrations of glycine to make the materials contain different proportions of brookite/anatase. Under visible light irradiation, the degradation rate of pure $TiO_2$ to ibuprofen was only 11.1%, but the degradation rate of C–N–$TiO_2$ was 100%.

Zhang et al. [320] synthesized a TiO$_2$ photocatalyst co-doped with fluorine and graphene oxide by the hydrothermal method. The decomposition rate of bromate by the complex TiO$_2$ photocatalyst was found to exceed 90% after UV irradiation for 15 min. The doping of fluoride ions can increase the number of high surface energy (001) crystal planes of TiO$_2$, while doped graphene oxide gives TiO$_2$ excellent conductivity and large specific surface area. Therefore, after introducing fluoride ions and graphene oxide, the transfer efficiency of photogenerated electrons can be effectively accelerated, and the photocatalytic efficiency can be improved.

### 5.3.3. Mixed Doping

Nonmetallic ions can also be used in combination with metal ions. Theoretically, the simultaneous doping of metal ions and non-metal ions can simultaneously obtain the advantages of metal ions and non-metal ions. Lei et al. [321] applied the hydrothermal method to prepare Fe, N, and C co-doped TiO$_2$ photocatalysts. In the study of the reduction and degradation of a Cr$^{6+}$ solution, the conversion of the Fe–N–C–TiO$_2$ catalyst reached 100%, which was much higher than other photocatalysts such as C–TiO$_2$ and N–C–TiO$_2$. Chen et al. [322] prepared TiO$_2$ co-doped with Ce and N via a sol–gel method and then incorporated diatomaceous earth particles into Ce–N–TiO$_2$ to form a composite photocatalyst. N ion doping reduces the forbidden bandgap of the TiO$_2$ composite catalyst, and the composite TiO$_2$ becomes a photoelectron trap due to the special 4f energy level of the Ce element. Therefore, under visible light irradiation for 240 min, the degradation rate of oxytetracycline by Ce–N–TiO$_2$/G is almost 100%, which is much higher than other catalysts. Han et al. [323] used a sol–gel method to prepare Zr, Ni and N co-doped TiO$_2$ catalyst. Under visible light irradiation, the conversion of NO and SO$_2$ gas by the three element co-doped TiO$_2$ photocatalyst is much higher than that of the single or two element (Zr, Ni, N) doped TiO$_2$ photocatalyst. Mixed doping can effectively increase the light absorption of the composite in the visible region, which is the main reason for the enhanced photocatalytic activity.

### *5.4. Surface Dye Photosensitization*

Surface dye photosensitization refers to the process of using a photosensitizer to broaden the wavelength response range of the photochemical reaction. Photosensitizers are usually some inorganic or organic chromophores. Photosensitizers have strong absorption under visible light and can extend the spectral response range of the photocatalytic system to the visible region. Therefore, the sensitized semiconductor catalyst can be excited with visible light of a much lower energy than ultraviolet light, which can broaden the application of a photocatalyst in the visible light region. Previous work on the photosensitization of semiconductor dyes mainly involved solar cells and photocatalytic hydrogen production, and in recent years the use of dye-sensitized semiconductor catalysts for decomposing organic pollutants has attracted the intense attention of researchers. Surface dye photosensitization has become one of the common methods for modifying semiconductor catalysts. Common photosensitizers include various organic dyes and transition metal complexes such as Ru and Pt chloride.

Wan et al. [324] used a modified hydrothermal reflux composite preparation method to prepare porphyrin-sensitized TiO$_2$/reduced graphene oxide (TiO$_2$/RGO) composite nanorod catalysts. Under visible light irradiation, the decomposition rate of methylene blue by TiO$_2$/RGO composite nanorods is very high at 92%, which is 4.3 times the decomposition rate of pure TiO$_2$ nanorods to methylene blue. Zoltan et al. [325] used three asymmetric sensitizers 5-(p-Nitrophenyl)-10,15,20-triphenylporphyrin, Cu(II)-porphyrin, and Zn(II)-porphyrin] for the sensitization modification of polycrystalline TiO$_2$ powder. The experimental results showed that the degradation rate of Congo red by the Zn(II)-porphyrin sensitized TiO$_2$ semiconductor catalyst is higher than that of other photocatalysts. Zhao et al. [326] attached the carboxylate porphyrins of Cu, Co, and Zn to the surface of TiO$_2$ by a hydrothermal method. In the degradation study of nitrophenol, the three sensitized semiconductor catalysts had a higher catalytic activity than pure TiO$_2$. Among the three sensitized semiconductor catalysts, Cu-porphyrin-TiO$_2$ has the best decomposition ability of nitrophenol. This is because Cu ions are excellent electron acceptors, and it is easier to attract electrons to reach a stable state, thereby achieving

the effect of separating electron-hole pairs. Wei et al. [327] used a two-step method of hydrothermal and reflux to prepare a Cu(II)-tetrakis(4-carboxyphenyl)porphyrin and RGO co-sensitized $TiO_2$ nanorod photocatalyst. After irradiation for 120 min under visible light, the decomposition rate of methylene blue by the composite $TiO_2$ photocatalyst was 95%, which was five times that of pure $TiO_2$ to methylene blue. Furthermore, the composite $TiO_2$ photocatalyst still maintained stable catalytic performance after being used six times in photocatalyst experiments.

Chowdhury et al. [328] reported the use of eosin dye to photosensitize $TiO_2$ and successfully reduced the band gap of a $TiO_2$ semiconductor by 1 eV. After sensitization, the catalyst showed a strong photo response in the visible region. Eosin dyes can inject excited electrons into the $TiO_2$ conduction band to achieve the same function as photogenerated electrons. Altin et al. [329] photosensitized Co-$TiO_2$ nano-doped particles with a metal-free complex-free phthalocyanine dye derivative by the sol–gel method. The prepared semiconductor catalyst exhibited an obviously red shift in the absorption wavelength, a shift which improved the application of the catalyst in the visible region. In the study of pollutant degradation, it was found that the photosensitization photocatalyst prepared by using Tween 20 surfactant had the best degradation performance, and the degradation rate of methyl orange (MB) dye (10 mg/L) by the photosensitization photocatalyst reached 60.3% under 150 min of visible light irradiation. Albay et al. [330] employed a deposition method to cure the novel Cu(II)-phthalocyanine derivative on the surface of $TiO_2$ nanoparticles to achieve a photosensitization modification. The results showed that photosensitization has little effect on the morphology of $TiO_2$, but photosensitized $TiO_2$ enhances the response to visible light. Under visible light, the bactericidal capacity of photosensitized $TiO_2$ and the conversion rate of photosensitized $TiO_2$ to $Cr^{6+}$ ions are much higher than those of a pure $TiO_2$ photocatalyst.

As a traditional modification method, surface dye photosensitization has been well developed in its development and application. One frequently used method of sensitization is to apply many dyes to the surface of the semiconductor. This method achieves good results in sensitization applications for nanoparticle solar cells, and there are articles about this method in famous journals such as *Nature* [331,332]. While the photosensitization of surface dyes can improve the photo responsive range and photocatalytic efficiency of photosensitized semiconductor catalysts, the photosensitization of surface dyes also has some drawbacks. First, the presence of surface photosensitizers limits the increase in photocatalytic efficiency of semiconductor photocatalysts. Second, the stability of photosensitizers for long-term use is questioned. Photosensitizers can fall off from the surface of the catalyst and can result in the secondary contamination to water. Furthermore, some photosensitizing dyes, such as ruthenium-based dyes, are expensive and not suitable for a wide range of applications.

In Table 3, we summarize both the advantages and disadvantages of the above modification methods.

**Table 3.** Advantages and disadvantages of different modification methods for semiconductor.

| Modification Method | Advantage | Disadvantage |
| --- | --- | --- |
| Particle doping | - Reduce bandgap<br>- Reduce particle size | - Introduce defects |
| Precious metal depositing | - Enhance electron-hole separation | - Expensive |
| Surface dye photosensitization | - Broaden the light response range | - Expensive<br>- Dyes may be photolyzed |
| Semiconductor compound | - Reducing the complex of electron-hole pairs<br>- Broaden the light response range | - Energy loss |

## 6. Application

### 6.1. Photocatalytic Hydrogen Production

In recent decades, the use of petroleum and coal resources has been increasing, and the number of unrepairable resources on Earth is getting fewer and fewer. Furthermore, combustion of fossil fuels has

caused many environmental problems that have significantly influenced the development of science and technology, as well as the daily life of humans.

Hydrogen is considered the most promising clean energy source of the 21st century. Hydrogen is characterized by being odorless, non-toxic, environmentally friendly, and capable of producing a large amount of energy. Hydrogen energy is a secondary energy source and needs to be prepared from other hydrogen-containing substances such as water and fossil fuels. There are a variety of ways of producing hydrogen, and the most common way is to decompose fossil fuels to produce hydrogen. However, this method relies on fossil fuels in the final analysis. The photocatalytic degradation of water to produce hydrogen to achieve the conversion of solar energy to hydrogen energy is an effective method to solve energy and environmental problems. The photocatalytic decomposition of water to generate hydrogen can convert solar energy into hydrogen energy, which is an effective method to solve energy and environmental problems.

Since the first photocatalytic decomposition of water to produce hydrogen in 1972, researchers have used powdered semiconductor materials to achieve the photocatalytic decomposition of water to produce hydrogen [7,215? –220]. Photocatalytic hydrogen production not only achieves the use of solar energy but also reduces environmental problems caused by energy use. Therefore, photocatalytic technology is considered to be the most ideal and cleanest way to produce and utilize energy [333,334].

Monoatomic catalysts can reduce costs by reducing the amount of precious metals used, but the challenge is how to maintain catalyst stability. Zhang et al. synthesized a bimetallic MXene nanosheet $Mo_2TiC_2T_x$ [335]. A large amount of Mo vacancies formed in the outer layer of the $Mo_2TiC_2T_x$, and after the Pt atoms were fixed in these vacancies, the hydrogen generation effectiveness of the $Mo_2TiC_2T_x$ was greatly improved, as shown in Figure 8B. The hydrogen evolution catalytic activity of the catalyst reached a current density of 100 mA/cm$^2$, requiring only an overpotential of 77 mV. $Mo_2TiC_2T_x$ has a 40-fold increase in mass activity compared to the commercially available platinum–carbon catalyst. Studies have shown that the superior catalytic performance and stability of $Mo_2TiC_2T_x$ arises from the covalent effect of Pt and MXene. As shown in Figure 8A, $Mo_2TiC_2T_x$-PtSA showed the best performance in all samples.

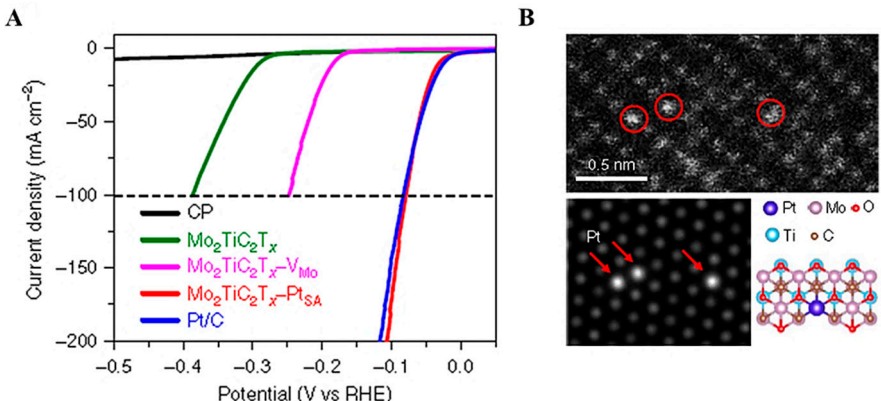

**Figure 8.** (**A**) Electrocatalytic performance for $Mo_2TiC_2T_x$-PtSA and reference HeR catalysts. (**B**) Magnified HAADF–STEM image of $Mo_2TiC_2T_x$-PtSA and its corresponding simulated image, as well as an illustration of the structure of $Mo_2TiC_2T_x$-PtSA, showing the isolated Pt atoms (circles in c) [335].

Bi et al. [336] synthesized g-$C_3N_4$ combined with PtNi$_x$. As shown in Figure 9A, the PtNi$_x$/g-$C_3N_4$ composites with different mass ratios have higher photocatalytic activities than pure g-$C_3N_4$. In the composite material, 2.5% PtNi$_x$/g-$C_3N_4$ (8456 µmol·h$^{-1}$g$^{-1}$) shows the highest rate of photocatalytic hydrogen generation, which is about 16 times of the rate of pure g-$C_3N_4$ (515 µmol·h$^{-1}$g$^{-1}$) photocatalytic hydrogen. From Figure 9B, the emission peak of pure g-$C_3N_4$ at about 470 nm can be observed, which is recombined in response to band gap electron-hole pairs. In addition, 2.5% PtNix/g-$C_3N_4$ shows a weaker emission peak intensity relative to pure g-C3N4, which means that the photoexcitation electron

and hole recombination rate in 2.5% PtNi$_x$/g-C$_3$N$_4$ is lower [337,338]. In other words, after loading PtNi$_x$ onto g-C$_3$N$_4$, it can effectively suppress the recombination of the photogenerated charge, thereby improving photocatalytic performance.

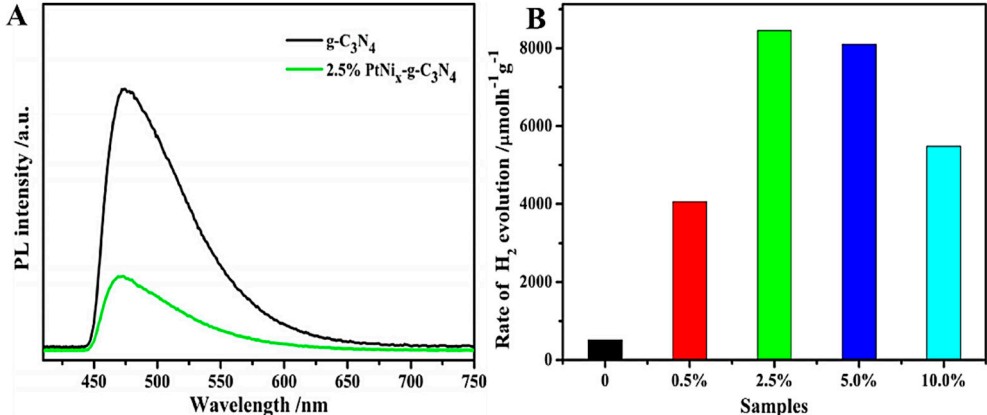

**Figure 9.** (**A**) Comparison of the photocatalytic H$_2$ production activity of g-C$_3$N$_4$, 0.5% PtNi$_x$/g-C$_3$N$_4$, 2.5% PtNi$_x$/g-C$_3$N$_4$, 5.0% PtNi$_x$/g-C$_3$N$_4$, and 10.0% PtNi$_x$/g-C$_3$N$_4$ samples using triethanolamine as the scavenger under a 500 W Xe lamp irradiation. (**B**) Photoluminescence (PL) spectra of 2.5% PtNi$_x$-g-C$_3$N$_4$ and g-C$_3$N$_4$ [336].

Jiang et al. [339] prepared a Bi$_3$TiNbO$_9$ and Cr/Nb co-doped Bi$_3$Ti$_{1-2x}$Cr$_x$Nb$_{1+x}$O$_9$ (x = 0.1, 0.2, 0.3) photocatalyst by the solid state reaction method. The doping of Cr/Nb can reduce the band gap of Bi$_3$TiNbO$_9$ by 1 eV. Figure 10A shows the hydrogen evolution experiment of all samples under full range illumination. Continuous hydrogen generation was detected through the experiment, and Cr/Nb co-doped Bi$_3$TiNbO$_9$ showed a higher hydrogen generation rate than pure Bi$_3$TiNbO$_9$. In the doped sample, Bi$_3$Ti$_{0.8}$Cr$_{0.1}$Nb$_{1.1}$O$_9$ (x = 0.1) showed the highest hydrogen generation rate, which was about twice that of pure Bi$_3$TiNbO$_9$. However, as the amount of Cr/Nb doping further increased, the hydrogen generation rate of the photocatalyst after doping decreased by 40%. The average photocatalytic hydrogen production rate under different irradiation conditions is given in Figure 10B.

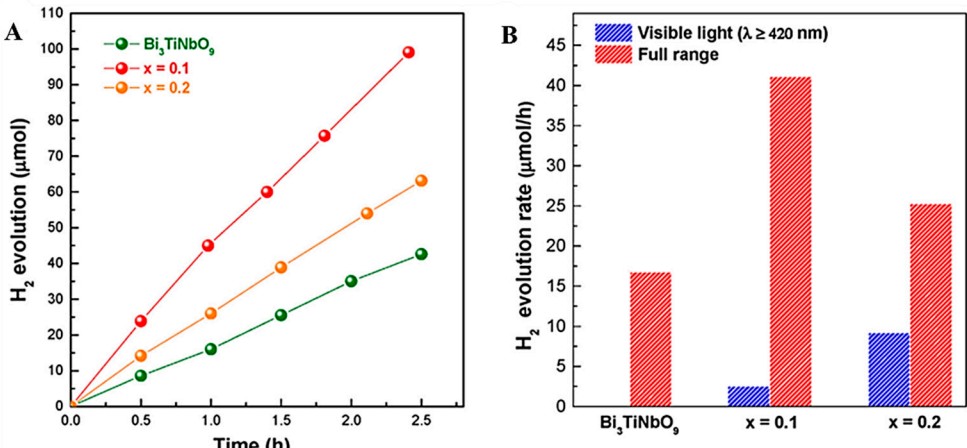

**Figure 10.** (**A**) Photocatalytic hydrogen production of all samples under full range illumination (λ ≥ 250 nm) in sodium sulfite aqueous solution (0.05 M). (**B**) Average photocatalytic hydrogen production rate under full range (λ ≥ 250 nm) and visible light illumination (λ ≥ 420 nm) [339].

*6.2. Wastewater Treatment*

Since the beginning of the 21st century, the problem of water pollution has become increasingly severe, and it has become a major issue affecting human health that restricts the harmonious

development of society. One of the main symptoms of water pollution is industrial wastewater. Industrial wastewater often includes oil substances, reaction aids, paper stocks, acids and alkaline substances, fiber impurities, dyes, and inorganic salt substances. Industrial wastewater is generally difficult to degrade. Today, methods for decomposing industrial wastewater are mainly biological methods, chemical methods, and physical methods. However, the above three methods do not clean the waste water completely—they all have some drawbacks.

In 1977, Frank et al. applied photocatalytic technology to the degradation of pollutants in water systems [340], laying the foundation for the use of photocatalytic oxidation technology in pollutant treatment. In the photocatalytic degradation of water pollution, active substances such as $H^+$, $H_2O_2$, and $\cdot OH$ generated by photocatalysts have a strong oxidizing activity and can deeply oxidize most of the organic pollutants in water into harmless small molecules. Therefore, a photocatalyt can be used for the purification of wastewater containing organic pollutants. Photocatalysis technology as a new high-efficiency and energy-saving modern sewage treatment technology has many advantages in wastewater treatment [341–352]. Compared to the traditional wastewater treatment method, photocatalytic technology can fundamentally remove pollutants, and, regardless of the pollutants' liquid or gaseous state, photocatalytic technology can have a good degradation effect.

In 2001, Houas et al. [351] used methylene blue to simulate environmental wastewater. Their experimental results showed that titanium dioxide based photocatalysts can successfully degrade methylene blue. Lachheb et al. [352] used UV light to irradiate a titanium dioxide photocatalyst to achieve the degradation of five different dyes (alizarin S, crocein orange G, methyl red, Congo red, and methylene blue). Experimental results showed that the five dyes not only decolorized successfully but also completely decomposed. Liu et al. [174] prepared flexible composite $Fe_2O_3/TiO_2$ nanofibers by combining a traditional electrospinning method and calcination process, which they successfully applied to photocatalytic wastewater treatment and photocurrent detection. In the photocatalytic degradation of rhodamine B experiment, $Fe_2O_3/TiO_2$ nanofibers have higher photocatalytic activity than pure $TiO_2$ under UV and visible light. The most important thing is that the $Fe_2O_3/TiO_2$ photocatalyst can be recovered by magnetic materials, thereby avoiding secondary pollution to the environment after photocatalytic treatment. Zhang et al. [167] synthesized flexible rime-like branched $TiO_2/PVDF$ composites via the electrospinning method and the hydrothermal method. In the photolysis rhodamine B experiments, the prepared flexible $TiO_2/PVDF$ composites showed a higher photocatalytic efficiency than pure PVDF. In particular, the $TiO_2/PVDF(H10)$ composite treated for 10 h in the hydrothermal reaction showed the highest photocatalytic activity (as show in Figure 11A). As shown in Figure 11B, the $TiO_2/PVDF(H10)$ composite still maintains its high photocatalytic efficiency after five photocatalytic experiments.

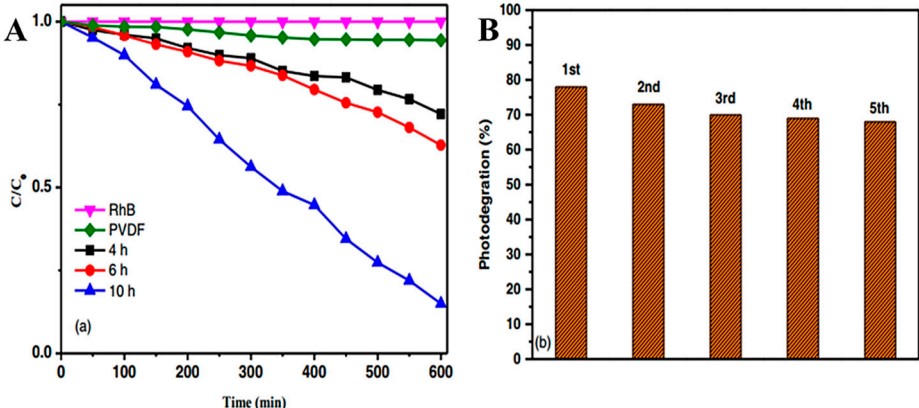

**Figure 11.** (**A**) Photodegradation profiles of rhodamine B over the samples: Rhodamine B without photocatalyst, PVDF and $TiO_2/PVDF$ composites hydrothermal treated for 4 h, 6 h, and 10 h. (**B**) The photocatalyst treated for 10 h was used for rhodamine B degradation, and the number of cycles was five times [167].

Regmi et al. [353] synthesized Ni-doped $BiVO_4$ photocatalysts using the microwave hydrothermal method. In the photocatalytic degradation experiments, the Ni-doped $BiVO_4$ photocatalyst showed better photocatalytic performance than pure $BiVO_4$. As shown in Figure 12, pure $BiVO_4$ has a strong absorption in the UV to 510 nm range, while the absorption edge of Ni-doped samples is significantly red-shifted as the Ni dopant concentration increases. In Figure 13, Regmi and coworkers evaluated the photocatalytic disinfection activity of $BiVO_4$ and Ni-doped $BiVO_4$ by inactivating *Escherichia coli* in an aqueous buffer solution under different light sources. The results show that pure $BiVO_4$ and 1 wt% Ni-doped $BiVO_4$ require only 1 h to inactivate *E. coli* to 100% under full spectrum sunlight illumination. However, after visible light irradiation for 5 h, the inactivation rate of pure $BiVO_4$ against *E. coli* reaches 72%, while the 3 wt% and 1 wt% Ni-doped $BiVO_4$ against *E. coli* inactivation rate of is 89% and 92%, respectively. Obviously, 1 wt% Ni-doped $BiVO_4$ shows higher efficiency, and as the doping amount of Ni increases the inactivation rate of *E. coli* gradually decreases.

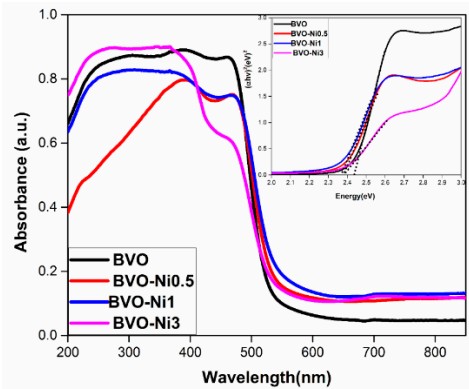

**Figure 12.** UV-Vis diffuse reflectance spectra of $BiVO_4$ samples with different wt% of Ni ions. Inset is the Kubelka–Munk plot with the corresponding band gap energies [353].

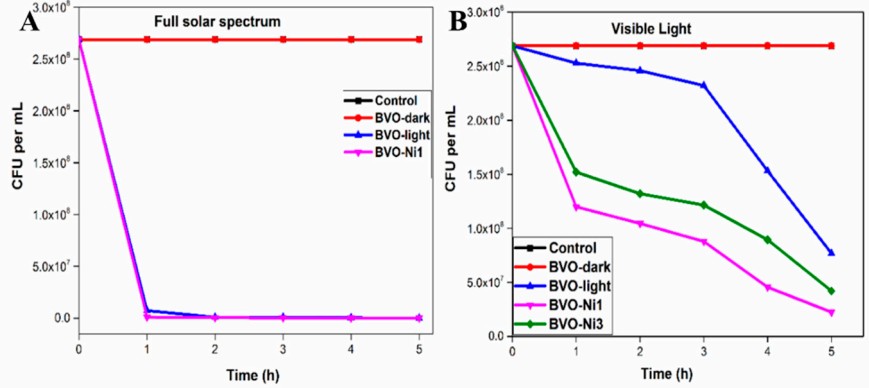

**Figure 13.** Inactivation of *E. coli* on: (**A**) Irradiation to full solar spectrum light; (**B**) visible light [353].

Cai et al. [354,355] developed a simple two-step method to prepare three-dimensional ordered macroporous (3DOM) gold-loaded $TiO_2$ photonic crystal photocatalysts. Based on this method, following the procedures shown in Figure 14A, i-$TiO_2$-o and i-Au–$TiO_2$-o photocatalysts were further generated. Both i-$TiO_2$-o and i-Au–$TiO_2$-o films prepared by this method exhibit a highly ordered interconnected porous structure, which is advantageous for the adsorption of target contaminants and utilization of solar energy. In the decomposition experiments of benzoic acid (BA), both i-$TiO_2$-o and i-Au–$TiO_2$-o showed higher decomposition rates than anatase $TiO_2$. As shown in Figure 14B, under ultraviolet irradiation, the decomposition rate of BA by i-Au–$TiO_2$-o is five or more times the decomposition rate of BA by anatase-type $TiO_2$.

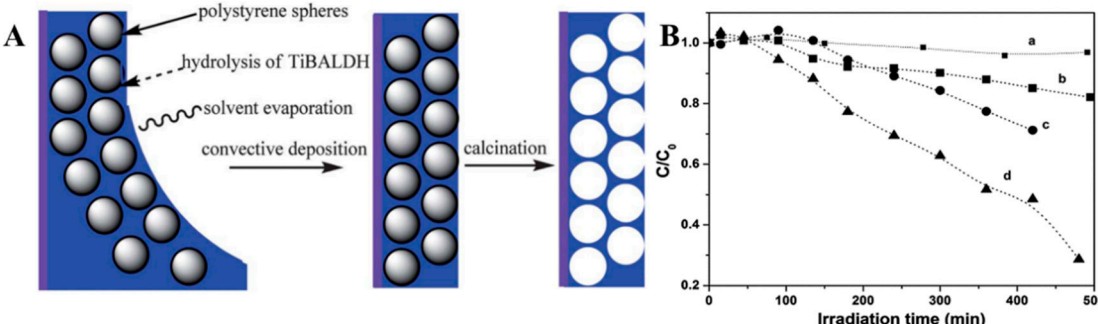

**Figure 14.** (**A**) Schematic illustration of procedures for the fabrication of 3DOM $TiO_2$ films loaded with gold nanoparticles. (**B**) Photocatalytic degradation of benzoic acid (BA) over (a) i-Au–$TiO_2$-o without UV irradiation, (b) nanocrystalline $TiO_2$, (c) i-$TiO_2$-o, and (d) i-Au–$TiO_2$-o under UV irradiation [354].

*6.3. Photocatalytic Disinfection*

As well as non-biological contaminants such as organic pesticides, antibiotics, and heavy metals in water, the harm caused by biological pollution such as medical sewage and domestic wastewater cannot be underestimated. Most pathogenic microorganisms can cause harm to human health by ingestion, respiration, or skin contact. According to the report, millions of people are dying or dead as a result of infection with pathogens annually [356].

Traditional sterilization methods for disinfection such as chlorination, ozonation, and ultraviolet light all have disadvantages. Chlorine can kill most bacteria and viruses, but the chlorine byproducts generated during the reaction can also endanger human health. Ozone oxidation technology is highly demanding on equipment and complicated to operate. Low-pressure ultraviolet disinfection can only inactivate bacteria on the surface of sewage; at the same time, the equipment is poor in reliability and expensive. For the first time, Matsunaga et al. [357] demonstrated that photocatalytic technology can eliminate pathogens including *Lactobacillus*, yeast, and *Escherichia coli*. Thereafter, photocatalytic antibacterial materials have been gradually known and developed.

In the past few decades, the photocatalytic technology of nanomaterials has been rapidly developed in solar photocatalytic sterilization and other aspects [358–367]. The reaction mechanism of photocatalytic sterilization is to destroy the cell wall and oxidize coenzyme A enzymes and genetic material. Under ultraviolet light irradiation, ·OH radicals and $O^{2-}$ radicals generated on the surface of the photocatalyst easily adhere to the cell wall surface of the bacteria. These free radicals can cause the leakage of $K^+$ ions in the cell fluid, the oxidation of coenzyme A in the cell, or the destruction of the DNA double helix structure. There is also a class of photocatalysts that do not produce active ·OH during photocatalytic and antimicrobial processes, and this type of photocatalyst sterilizes pathogens using photogenerated holes. As a new sterilization method, photocatalytic sterilization has the advantages of high safety, strong stability, and wide antibacterial properties, and it demonstrates broad application prospects.

Obuchi et al. [368] obtained $TiO_2/SiO_2$ and Ag–$TiO_2/SiO_2$ photocatalysts by hydrolysis and the calcination method. Figure 15 shows the survival rate of spore-forming *Bacillus subtilis* to treatment time under UV irradiation and dark conditions and the relationship between the doping amount of Ag and the disinfection rate constant, respectively. Figure 15A shows that the viability of sporulated *Bacillus subtilis* reduced only when using Ag–$TiO_2/SiO_2$ under ultraviolet irradiation. As shown in Figure 15B, the optimum range of the Ag doping amount is 0.5–1.0 wt%.

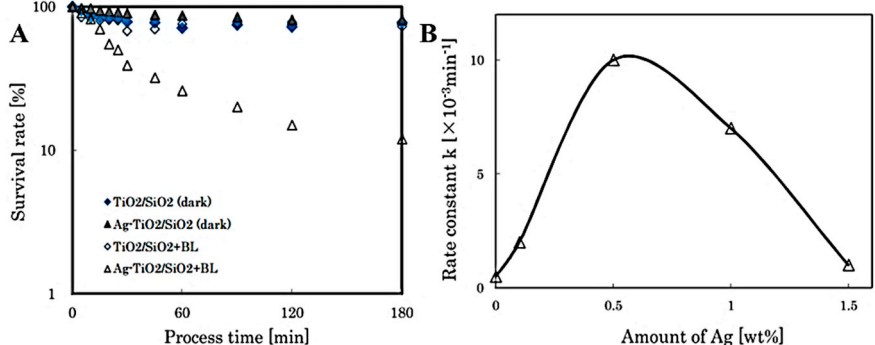

**Figure 15.** (**A**) Survival rate of sporulating *Bacillus subtilis* against process time using TiO$_2$/SiO$_2$ and Ag-doped TiO$_2$/SiO$_2$ (BL; under ultraviolet irradiation, dark; without UV irradiation). (**B**) Relationship between Ag amount and rate constant [368].

### 6.4. Air Purification

Air quality is closely related to the health of human beings. In recent years, researchers have agreed that air pollution is closely related to respiratory infections, lung diseases, cardiovascular disease, and coronary heart disease [68–73,369–372]. To date, the main sources of air pollution are industrial waste gas and automobile exhaust gas. Sulfur compounds such as sulfur dioxide, nitrogen oxides, halogen-containing compounds, and malodorous gases contained in industrial exhaust gases, as well as carbon monoxide, nitrogen oxides, and sulfur oxygen compounds in automotive exhaust gases, have a significant impact on people's health. Another major pollutant is volatile organic compounds (VOCs). VOCs are organic pollutants widely found in indoor and outdoor air. VOCs are precursors of PM 2.5 and also are the most important air pollutants after PM 2.5 [373,374].

With the development of photocatalysis, the use of photocatalysts to decompose air pollutants has received increasing attention from researchers [375–382]. Photocatalytic material could adsorb or decompose toxic gases in the atmosphere, reducing the adverse effects of toxic gases on the environment. Semiconductor photocatalysis is mild, and the reaction process is relatively simple. In theory, photocatalysis can degrade almost all air pollutants. Therefore, compared with conventional methods such as filtration, adsorption, plasma, and ozone oxidation, photocatalytic technology can completely degrade the pollutants in the air under sunshine, thereby rapidly purifying the air.

Photocatalysis is a technology with wide applications and great development potential. As shown in Figure 16, in addition to the above four applications, photocatalytic technology is also widely used in agriculture, construction, automobiles, roads, and household appliances.

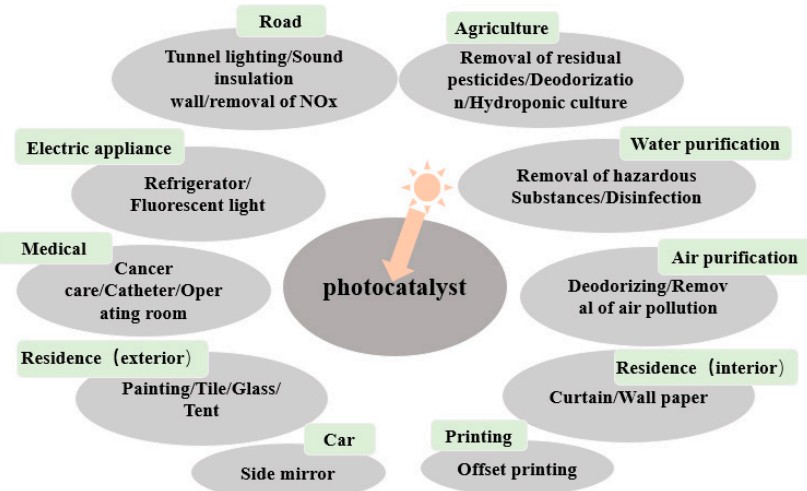

**Figure 16.** Applications of semiconductor photocatalysis.

## 7. Summary and Outlook

In summary, the study on photocatalytic technology has developed for decades and has achieved substantial breakthrough in theory and a series of practical application results. Due to the simple photocatalysis oxidation equipment, easy control of operating conditions, strong oxidizing ability, no secondary pollution, and its broad applications in photocatalytic hydrogen production, the degradation of sewage, sterilization, and the purification of air, it has become a highly promising technology. In this review, the mechanism and application of photocatalyst technology, classification, modification, etc. have been summarized in detail. To date, the development of photocatalytic technology is far from meeting the demand for industrialization for many reasons. The photocatalyst prepared at this stage has several disadvantages such as the low utilization of sunlight, low quantum efficiency, low recyclability, and low photocatalyst efficiency. Therefore, researchers still need to devote more efforts to find a highly efficient and stable visible light photocatalyst in order to improve photocatalytic efficiency and promote the application of photocatalytic technology.

**Acknowledgments:** This work was supported by the National Natural Science Foundation of China (51,673,103) and the National Natural Science Foundation of China (51,703,102). Z.C. is grateful for the support from the joint French-Singaporean MERLION program (Grant No. R279,000,334,133).

**Funding:** This research was funded by the National Natural Science Foundation of China (51,703,102).

**Conflicts of Interest:** The authors declare no conflict of interest.

## Abbreviations

| Full Name | Abbreviation |
|---|---|
| Valence band | VB |
| Conduction band | CB |
| Direct-current power supply | DC power supply |
| Physical vapor deposition | PVD |
| Chemical vapor deposition | CVD |
| Molecular beam epitaxy | MBE |
| RF magnetron sputtering | RF-MS |
| Ultraviolet irradiation | UV irradiation |
| Methyl orange | MB |
| Rhodamine B | RhB |
| *Escherichia coli* | *E. coli* |
| Benzoic acid | BA |
| Volatile organic compounds | VOCs |

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
