# Peer review of "Recent Advances and Applications of Semiconductor Photocatalytic Technology"

_applsci, doi:10.3390/app9122489_

Round 1

Reviewer 1 Report

General Comments

The manuscript presents a review of photocatalytic technologies, reactions, the influencing factors, and types of photocatalysts. Common synthesis methodologies are summarized together with some applications of photocatalysis. Overall, this could become a potential important review, but to be complete some missing information about photocatalysts that have not been mentioned yet and their applications should be included in the manuscript. For example, among the sulfides included, in addition to CdS and CuS, it would be really important that the manuscript includes ZnS and all of the following 14 key references:

1)      EGGINS, BR; ROBERTSON, PKJ; STEWART, JH; et al., JOURNAL OF THE CHEMICAL SOCIETY-CHEMICAL COMMUNICATIONS  Issue: 4   Pages: 349-350   Published: FEB 21 1993.

2)      Eggins, BR; Robertson, PKJ; Murphy, EP; et al., JOURNAL OF PHOTOCHEMISTRY AND PHOTOBIOLOGY A-CHEMISTRY  Volume: 118   Issue: 1   Pages: 31-40   Published: OCT 15 1998

3)      HENGLEIN, A; GUTIERREZ, M; FISCHER, CH BERICHTE DER BUNSEN-GESELLSCHAFT-PHYSICAL CHEMISTRY CHEMICAL PHYSICS  Volume: 88   Issue: 2   Pages: 170-175   Published: 1984.

4)      Johne, P; Kisch, H, JOURNAL OF PHOTOCHEMISTRY AND PHOTOBIOLOGY A-CHEMISTRY  Volume: 111   Issue: 1-3   Pages: 223-228   Published: DEC 1997

5)      KANEMOTO, M; SHIRAGAMI, T; PAC, CJ; et al., JOURNAL OF PHYSICAL CHEMISTRY  Volume: 96   Issue: 8   Pages: 3521-3526   Published: APR 16 1992

6)      Zhang, Xiang V.; Ellery, Shelby P.; Friend, Cynthia M.; et al., JOURNAL OF PHOTOCHEMISTRY AND PHOTOBIOLOGY A-CHEMISTRY   Volume: 185   Issue: 2-3   Pages: 301-311   Published: JAN 25 2007.

7)      Guzman, MI; Martin, ST., INTERNATIONAL JOURNAL OF ASTROBIOLOGY   Volume: 7   Issue: 3-4   Pages: 271-278   Published: JUL-OCT 2008.

8)      Guzman, MI; Martin, ST., ASTROBIOLOGY   Volume: 9   Issue: 9   Pages: 833-842   Published: NOV 2009.

9)      Zhang, Xiang V.; Martin, ST., JOURNAL OF THE AMERICAN CHEMICAL SOCIETY   Volume: 128   Issue: 50   Pages: 16032-16033   Published: DEC 20 2006

10)   Guzman, MI; Martin, ST, CHEMICAL COMMUNICATIONS   Volume: 46   Issue: 13   Pages: 2265-2267   Published: 2010.

11)   Zhou, Ruixin; JOURNAL OF PHYSICAL CHEMISTRY C   Volume: 118   Issue: 22   Pages: 11649-11656   Published: JUN 5 2014.

12)   Zhou, Ruixin; JOURNAL OF PHYSICAL CHEMISTRY C   Volume: 120   Issue: 13   Pages: 7349-7357   Published: APR 7 2016.

13)   Zhou, Ruixin; Basu, Kaustuv; Hartman, Hyman; et al., SCIENTIFIC REPORTS   Volume: 7     Article Number: 533   Published: APR 3 2017.

14)   Hoque, Md. Ariful, MATERIALS   Volume: 11   Issue: 10     Article Number: 1990   Published: OCT 2018

The example above could be applied to other interesting photocatalysts. The key will be that the authors of the review provide a concise and precise updated manuscript that shows the progress made in this research area following the PRISMA guidelines. Please note that PRISMA covers systematic reviews and the authors are recommended to complete the checklist and flow diagram and include it with their submission:

http://prisma-statement.org/

http://prisma-statement.org/PRISMAStatement/Checklist.aspx

http://prisma-statement.org/PRISMAStatement/FlowDiagram.aspx

Author Response

We thank the reviewer’s comments and added the content on ZnS as a new section. We now have cited all the papers that the reviewer recommended to the references. The revisions can be found on page 7, line 206.

Reviewer 2 Report

The submitted article is a review dealing with recent advances of photocatalytic processes. Numerous reviews on this subject have already been published. This one does not bring any further information and is not complete in terms of cited references.

Author Response

Regarding the novelty of this manuscript, a concern by reviewer #2, we added additional text to the manuscript. We comprehensively introduced photocatalyst technology, and reviewed the latest development in photocatalyst technology, which are not done by other review articles. In this review article, we also presented some new representative examples. The significance of our manuscript has been recognized by both Reviewer #1 and Reviewer #3. Reviewer #1 commented that “this could become a potential important review”. Reviewer #3 also pointed that “this work is interesting and presented with lot of supporting evidence”. It is well known that photocatalysis is a rapidly developing research topic. It is true that researchers have published many review articles. However, we found that many review articles only described for one or several aspects of photocatalytic technology. A comprehensive review on the photocatalytic technology is missing. In this review, we aim to provide a comprehensive and up-to-date summary on photocatalysis studies to researchers. Through this article, we hope that more researchers will understand photocatalyst technology and learn the progress in industrialization of photocatalyst technology. In addition, we have supplemented the references. We hope this revision can satisfy your requirement.

Reviewer 3 Report

The work is interesting and presented with lot of supporting evidence. In my opinion lot of typographical errors and mismatching of diagrams, captions and text. The author must be addressed the following comments and need careful revision of the entire manuscript.

1.      Abstract section is not clear. Write very concise and specifically.

2.      Introduction section needs a major revision. This section must be rewritten in a standard way.

3.      Lot of typographical errors and grammatical mistakes noticed in the manuscript. Need compete check for the entire manuscript. correct color of all reference for example 1-3, 74-87. Spacing between band 199 gap (2.7-2.8 eV) on page number 7 line 199, correct any one out of figure or fig. ,

4.      Correct all Figure caption is not the experimental setup - check the figure (some time bold .

5.      The main advantages and disadvantages of present study must be added.

6.      Provide the literature assessment and cite the studies, and mention very clearly the difference from previous studies and the novelty of this study.

7.      Please pay much attention to the significant digits in some tables.

8.      Summary and outlook section must be improved.

9.      Please add recent references in introduction part:

10.  Figure 11 and 13 are missing.

11.  All figures need to improved quality with journal formats.

12.  All paper should revised. Author need to major improvements in all section.

13.  Correct reference no. 10, 17, 22, 27, 41, 48, 51, 54 and all reference

Author Response

Comment #1, “Abstract section is not clear. Write very concise and specifically.” Response: We rewrote the abstract and tried to make it clear. Comment #2, “Introduction section needs a major revision. This section must be rewritten in a standard way.” Response: We rewrote Introduction section according to the Reviewer’s suggestion. We hope now the introduction can meet the requirement. Comment #3“Lot of typographical errors and grammatical mistakes noticed in the manuscript. Need compete check for the entire manuscript.” Response: We have corrected typographical errors and grammatical mistakes. These errors include Fig and spacing between band gap (2.7-2.8 eV). These errors have been corrected Figure 2 on page 5, line 120 and spacing between band gap (2.7-2.8 eV) on page 7, line 240. Comment #4 “Correct all Figure caption is not the experimental setup - check the figure.” Response: We thank the reviewer for the kind suggestions. We now have corrected all Figure captions. Comment #5 “The main advantages and disadvantages of present study must be added.” Response: Although some progress has been achieved in current photocatalytic research, for example, in the improvement of photocatalytic efficiency and absorption of visible light, the current photocatalyst technology is still insufficient for industrialization. There is still a need to find photocatalysts with higher photocatalytic efficiency in visible light, and can be recycled. Comment #6 “Provide the literature assessment and cite the studies, and mention very clearly the difference from previous studies and the novelty of this study.” Response: We thank the reviewer for the very nice suggestions. We now have provided the literature assessment and cite the studies, and mention very clearly the difference from previous studies and the novelty of this study. Comment #7 “Please pay much attention to the significant digits in some tables.” Response: We have revised the significant digits in some tables according to your requirements Comment #8 “Summary and outlook section must be improved.” Response: We carefully considered your suggestions and rewrote the Summary and Outlook section. This revision can be found on page 27, line 930. Comment #9 “Please add recent references in introduction part.” Response: We now have made more changes and cited more recent references. These changes include re-written Introduction section and add recent references in Introduction section. This revision can be found on page 2, line 34. Comment 10 “Figure 11 and 13 are missing.” Response: We now have added Figures 11 and 13. Comment #11 “All figures need to improved quality with journal formats.” Response: We have improved the quality of the figures. Comment #12 “All paper should revised. Author need to major improvements in all section.” Response: We have carefully improved the quality of the writing to make the paper more readable and more attractive to readers. Comment #13 “Correct reference no. 10, 17, 22, 27, 41, 48, 51, 54 and all reference.” Response: We have corrected the format of the above references and all other references.

Round 2

Reviewer 1 Report

After the revision the manuscript is recommended to be published as is.

Author Response

Thank you for your affirmation of our article, we will continue to work hard

Reviewer 2 Report

This work has been nicely revised according to the reviewers' comments. However, to my point of view, the part concerning applications of photocatalysis is still poor and the cited references are still not complete for a review that intends to be significant in the field for many researchers. The cited articles are mainly from Asia. Some papers written by European and American researchers are not cited (P. Pichat, C. Guillard, S. Lacombe, G. Li Puma, etc.). I suggest the authors to develop the part on applications or to restrain their article to novelty on photocatalysts development.

Author Response

We thank the reviewer’s comments and we have also cited papers written by researchers in Europe and the United States (P. Pichat, C. Guillard, S. Lacombe, G. Li Puma, etc.). We now have made more changes and cited more references. We also cite several photocatalytic papers, hoping to improve the integrity of the photocatalytic application part of this manuscript. In addition, we have supplemented the references. We hope this revision can satisfy your requirement.
